

# Four years of global carbon cycle observed from OCO-2 version 9 and *in situ* data, and comparison to OCO-2 v7

Hélène Peiro [1], Sean Crowell [1], Andrew Schuh [2], David F. Baker [2], Chris O'Dell [2], Andrew R. Jacobson [3,4], Frédéric Chevallier [5], Junjie Liu [6], Annmarie Eldering [6], David Crisp [6], Feng Deng [7], Brad Weir [8,9], Sourish Basu [10,11], Matthew S. Johnson [12], Sajeev Philip [13], and Ian Baker [14]

[1]University of Oklahoma, Norman, OK, USA
[2]Cooperative Institute for Research in the Atmosphere, Colorado State University, Fort Collins, CO, USA
[3]Cooperative Institute for Research in Environmental Sciences, University of Colorado Boulder, Boulder, CO, USA
[4]NOAA Global Monitoring Laboratory, Boulder, CO, USA
[5]Laboratoire des Sciences du Climat et de L'Environnement, LSCE/IPSL, CEA-CNRS-UVSQ,Université Paris-Saclay, 91198 Gif-sur-Yvette, France
[6]Jet Propulsion Laboratory, California Institute of Technology, Pasadena, CA, USA
[7]Department of Physics, University of Toronto, Toronto, Ontario, Canada
[8]Universities Space Research Association, Columbia, MD, USA
[9]NASA Goddard Space Flight Center, Greenbelt, MD, USA
[10]NASA Goddard Space Flight Center, Global Modeling and Assimilation Office, Greenbelt, MD, USA
[11]Earth System Science Interdisciplinary Center, College Park, MD, USA
[12]NASA Ames Research Center, Moffett Field, CA, USA
[13]Universities Space Research Association, Mountain View, CA
[14]Colorado State University, Atmospheric Sciences, Fort Collins, CO, USA

**Correspondence:** Helene Peiro (helene.peiro@ou.edu)

**Abstract.**

   The Orbiting Carbon Observatory 2 (OCO-2) satellite has been provided information to estimate carbon dioxide ($CO_2$) fluxes at global and regional scales since 2014 through the combination of $CO_2$ retrievals with top-down atmospheric inversion methods. Column average $CO_2$ dry air mole fraction retrievals has been constantly improved. A bias correction has been

applied in the OCO-2 version 9 retrievals compared to the previous OCO-2 version 7r improving data accuracy and coverage. We study an ensemble of ten atmospheric inversions all characterized by different transport models, data assimilation algorithm and prior fluxes using first OCO-2 v7 in 2015-2016 and then OCO-2 version 9 land observations for the longer period 2015-2018. Inversions assimilating *in situ* (IS) measurements have been also used to provide a baseline against which to compare the satellite-driven results. The times series at different scales (going from global to regional scales) of the models emissions

are analyzed and compared to each experiments using either OCO-2 or IS data. We then evaluate the inversion ensemble based on dataset from TCCON, aircraft, and in-situ observations, all independent from assimilated data. While we find a similar constraint of global total carbon emissions between the ensemble spread using IS and both OCO-2 retrievals, differences between the two retrieval versions appear over regional scales and particularly in tropical Africa. A difference in the carbon budget between v7 and v9 is found over this region which seems to show the impact of corrections applied in retrievals.



However, the lack of data in the tropics limits our conclusions and the estimation of carbon emissions over tropical Africa require further analysis.

## 1 Introduction

Understanding the global carbon cycle and how quickly the planet warms in response to human activities is becoming a global
priority. $CO_2$ is a key driver of global warming and its dynamics can be explored with a variety of $CO_2$ measurements. Ground based (*in situ*) data, while highly precise and accurate, are distributed very sparsely over the globe (Ciais et al., 2013). Space-based $CO_2$ retrievals, on the other hand, allow comprehensive spatial coverage across the globe, particularly over regions with few surface observations, such as the tropics. Furthermore, the number of satellites observing atmospheric $CO_2$ has rapidly grown over the past decade, e.g. Greenhouse Gases Observing Satellite (GOSAT/GOSAT2, Kuze et al. (2009); Nakajima et al.
(2012)) and the Orbiting Carbon Observatory (OCO-2/OCO-3, Crisp et al. (2017); Eldering et al. (2017)).

The rise in $CO_2$ concentration at a global scale has motivated the drive towards a better understanding of the global surface fluxes of carbon (World Meteorological Organisation, 2020). In order to understand the different processes involved in the carbon cycle, such as uptake or release of $CO_2$ by the oceans and the land biosphere, and hence be able to predict future climate change, we need accurate emissions estimates and an improved understanding of natural $CO_2$ emissions and uptakes. Top-
down atmospheric inversion approaches that couple atmospheric observations of $CO_2$ with chemistry (atmospheric) transport models (CTMs) have been widely used to estimate $CO_2$ fluxes (Ciais et al., 2010; Peylin et al., 2013; Basu et al., 2013; Wang et al., 2018; Crowell et al., 2019). This is in contrast to "bottom-up" methods, which often use a mechanistic understanding of the carbon-cycle, e.g. soil dynamics, photosynthesis, decomposition processes, and steady state ocean-atmospheric chemical exchange, to predict land and ocean-atmospheric exchange, and hence atmospheric $CO_2$ concentrations. While the mechanistic
underpinnings of these models is attractive, there is no guarantee that the resulting atmospheric exchange of $CO_2$ will bear any similarity to reality. By contrast, the "top-down" approach often uses a "bottom-up" model output as a starting guess and then optimizes atmospheric exchange to agree with atmospheric observations.

Formal uncertainties of top-down approaches can be attributed to the errors in the observations assimilated, i.e. "observation" errors, and to errors in the starting guess from mechanistic models. However, past studies have also shown that top-down
estimates can be sensitive to errors in the modeled atmospheric transport as well as in choices related to the optimization technique (Chevallier et al., 2010; Houweling et al., 2015; Basu et al., 2018; Schuh et al., 2019), uncertainties which are difficult if not impossible to characterize formally in any one atmospheric inversion scheme. This shortcoming was the motivation for the OCO-2 Model Inter-comparison Project (MIP), whose goal was to (1) study the impact of assimilating OCO-2 retrieval data into several atmospheric inversion models and (2) provide an overall ensemble spread of the model emissions charac-
terizing most sources of known uncertainty. In addition to its primary goal of assimilating OCO-2 retrievals, MIP modelers





also assimilate *in situ* data, which has a long and documented history (Enting and Newsam, 1990; Enting, 2002; Gurney et al., 2002; Rayner et al., 2014). In the first iteration of the MIP project (the "v7 MIP"), OCO-2 version 7r land observations were used and analyzed for the 2015-2016 period (Crowell et al., 2019). In that study, the authors found good agreement between *in situ* and satellite inversions at the global scale. However, differences appeared at smaller regional scales, particularly over

the tropics in areas such as Northern Africa, where stronger sources were observed with the OCO-2 inversions than with the *in situ* inversions. The authors concluded that the differences over the tropics, besides being due to better observability in a region with few *in situ* observations, could be due to the global perturbation from the 2015-2016 El Niño.

Previous inversion studies have shown the importance of using accurate and precise satellite retrievals for the $CO_2$ flux inversion, particularly at regional scales (Chevallier et al., 2005; Basu et al., 2013; Maksyutov et al., 2013; Chevallier et al.,

2014; Deng et al., 2014; Feng et al., 2016; Crisp et al., 2017). What may appear to be very small biases in the remote sensing retrieval of column averaged $CO_2$ ($XCO_2$) can have large effects on resulting $CO_2$ fluxes from inversions. In support of bias reduction, OCO-2 retrievals have been validated against Total Carbon Column Observing Network (TCCON) data and a precision of 1-2 ppm has been estimated, with geographic $CO_2$ biases of unknown magnitude possibly present at regional scales.

In this study, we want to quantify satellite-informed fluxes from the latest OCO-2 retrievals (v9) at the global and regional scales and contrast differences with the previous flux estimates based on OCO-2 v7r data (Crowell et al., 2019). In some sense, the point of our paper is to update Crowell et al. (2019) paper with the latest flux inversion results based on the longest and most recent set of *in situ* and satellite $XCO_2$. In particular, this study aims at evaluating whether : (i) there is some change in the MIP $CO_2$ fluxes using OCO-2 v9 as compared to OCO-2 v7 data; and (ii) if there are some differences, what would be the

implications in the carbon cycle community of using v9 regarding previous studies that have used v7 ?

The paper is structured as follows. The MIP design as well as data used in the inversions (i.e., the *in situ* data and OCO-2 v9 retrievals, as well as how v9 differs from v7) will be detailed in section 2. In this same section the independent data used for evaluation will be also presented. Section 3 presents the optimized fluxes estimated from the *in situ*, v7 and v9 inversions at global, latitudinal and regional scales. Evaluation using independent data will appear at the end of this section 3. Finally,

section 4 will discuss the results and findings.

## 2   Methodology and data set

### 2.1   MIP design

The MIP project, organized by the OCO-2 Science Team, is a collaboration of $CO_2$ modelers formed to study the impact of assimilating OCO-2 retrieval data into atmospheric inversion models. The project's goal is to create an ensemble of $CO_2$ surface

flux estimates to understand how flux estimates using OCO-2 retrievals and *in situ* measurements depend on (i) transport, (ii) data assimilation methodology, (iii) prior flux and associated errors and (iv) possible systematic errors in the OCO-2 retrievals, in particular across viewing modes, i.e. ocean glint (OG), land nadir (LN), and land glint (LG). The OCO-2 MIP philosophically mimics past projects such as RECCAP (REgional Carbon Cycle Assessment and Processes) and TRANSCOM





(The Atmospheric Tracer Transport Model Intercomparison Project) designed to analyze the uncertainty in inverse calculations
of the global carbon budget resulting from errors in simulated atmospheric transport. Table 1 gives summary information of
the different modeling systems and their transport models and configurations, while Table 2 gives the information of modelers
names and their respective institutions. The modelers used NASA's operational bias-corrected OCO-2 L2 Lite $XCO_2$ product
v9 (Kiel et al. (2019), https://daac.gsfc.nasa.gov) in the v9 version of the MIP. The OCO-2 v9 dataset has an improved bias
correction approach that results in reduced biases, particularly over areas of rough topography. While variations amongst
inversion systems are considered beneficial for the purpose of characterizing flux uncertainty, some configurations needed
to be standardized in order to avoid meaninglessly large differences in the ensemble spread. All inversion modelers were
instructed to assimilate OCO-2 data from September 6th 2014 through May 31th 2019 and to submit estimated fluxes from
January 1, 2015 through December 31th, 2018 (to allow the flux estimate some time to spin up and down on either end).
Fossil fuel emissions, which are typically not optimized in global top down studies (Peylin et al., 2013), were standardized for
the project. Similar to the experiments described in Crowell et al. (2019), all modelers assumed the same monthly fossil fuel
emissions from the Open-source Data Inventory for Anthropogenic $CO_2$ (ODIAC2019, Oda and Maksyutov (2011), Oda et al.
(2018)), modified with the TIMES diurnal and day-of-week scaling (Nassar et al., 2013).

Though fossil fuel emissions are fixed, all other prior flux estimates were chosen independently by each modeling group.
For instance, regarding fire emissions, most of models used the Global Fire Emission Database either version 3 (GFED3) or
version 4 (GFED4). GFED3 and GFED4 mainly differ on burned area where small fires are included in version 4 (Randerson
et al., 2012; Giglio et al., 2013). The added information of small fire burned area increase the burned area particularly over
agricultural and peat land regions (Van Der Werf et al., 2017).

## 2.2 OCO-2 retrievals

The NASA satellite OCO-2 was launched in July 2014 (Crisp et al., 2017; Eldering et al., 2017) and flies in a near-polar,
sun-synchronous orbit (so groundtracks are spaced more closely at high latitudes than mid-latitudes) at a 705 km altitude with
a local crossing time at the Equator between 13:21 and 13:30 local time. OCO-2 flies in the EOS Afternoon Constellation (A-
Train) and has a sixteen-day ground track repeat cycle that gives global $XCO_2$ coverage twice per month with approximately
150 km longitudinal offsets between nearby revisiting orbits. OCO-2 has a spectrometer measuring sunlight reflected by the
Earth and its atmosphere in three spectral bands : the oxygen A-band in the near-infrared (NIR) at 0.76 μm wavelength, and two
$CO_2$ spectral bands in the shortwave infrared (SWIR) at 1.6 and 2.1 μm. OCO-2 provides spatially dense data with a narrow
swath (no wider than 10 km) and with a spatial footprints of a few square kilometres (less than 1.25 km by 2.2 km projected
onto the surface). O'Dell et al. (2018) reported that the fine resolution of OCO-2 increased the number of cloud-free scenes. As
is known, clouds are difficult to model, so having more cloud-free scenes yields more successful retrievals with lower errors.

The OCO-2 sensor provides observations in three modes. Nadir retrievals are those in which the satellite is looking at the
earth directly below, i.e. at the sub-satellite point. These retrievals are only usable when the instrument is directly over land.
Glint retrievals are from measurements occurring when the instrument is pointed (usually off-nadir) toward the solar glint
spot. Glint is the primary mode for over-ocean retrievals, as the ocean surface is very dark in the SWIR spectral range, only

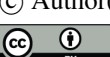



**Table 1.** Configuration of each simulations used in the MIP comparison.

| Simulation name | Transport model | Meteorology | Spatial resolution | Prior Land bio | Prior ocean | Prior fire | Inverse method |
|---|---|---|---|---|---|---|---|
| Ames | GEOS-Chem | MERRA-2 | 4x5 | CASA-GFED4.1s | CT2019OI | GFED4.1s | 4D-Var |
| CAMS | LMDz | ERA-Interim | 1.9x3.75 | ORCHIDEE | CMEMS | GFEDv4 | Variational |
| CMS-Flux | GEOS-Chem | GEOS-FP | 4x5 | CARDAMOM | ECCOS-Darwin | GFED4.1s | 4D-Var |
| CSU | GEOS-Chem | MERRA-2 | 1x1 | SIB4/ MERRA-2 | CT2019 OI | GFED4 | Bayesian synthesis |
| CT | TM5 | ERA-Interim | 3x2/ 1x1 | CT2019 CASA GFED4.1s | CT2019 OIF | CT2019 CASA-GFED4.1s | EnKF |
| OU | TM5 | ERA-Interim | 4x6 | CASA-GFED3 | Takahashi | GFEDv3 | 4D-Var |
| Baker | PCTM | MERRA-2 | 6.7x6.7 | CASA-GFED3 | Takahashi | GFEDv3 | 4D-Var |
| TM5-4DVAR | TM5 | ERA-Interim | 2x3 | SIB-CASA | CT2019 Opt Clim | GFEDv4 | 4D-Var |
| UT | GEOS-Chem | GEOS-FP | 4x5 | BEPS (Chen et al., 2012) | Takahashi et al. (2009) | GFEDv4 | 4D-Var |
| LoFI [a] | GEOS GCM | MERRA-2 | 0.5x0.625 | CASA-GFED3 | LoFI Takahashi | QFED | |

[a] This simulation has been used in the v9 MIP. LoFI uses a different method than the other inversions but fit some independent data as the other simulation do. In addition, it has the highest resolution. LoFI has then been used in this MIP project to look at a range of different methods but, in this study, we will focus our analyze to the range of emissions from the other simulations.



**Table 2.** Contact and institution of each simulation.

| Simulation name | Contact | Institution |
| --- | --- | --- |
| Ames | Matthew Johnson | NASA Ames Research Center |
| CAMS | Frédéric Chevallier | LSCE France |
| CMS-Flux | Junjie Liu | NASA JPL |
| CSU | Andrew Schuh | Colorado State University |
| CT | Andy Jacobson | University of Colorado and NOAA GML |
| OU | Sean Crowell | University of Oklahoma |
| Baker | David Baker | Colorado State University |
| TM5-4DVAR | Sourish Basu | University of Maryland and NASA GMAO |
| UT | Feng Deng | University of Toronto |
| LoFI | Brad Weir | NASA Goddard |

reflecting sufficient solar radiation near the glint point. Target mode retrievals are obtained when the sensor points at a fixed location along the orbit to keep a particular point on the Earth's surface in view, and is employed mainly to collect validation data over locations such as TCCON sites. In this study, besides using land nadir and land glint separately from the v7 MIP, both land nadir-and land glint-mode retrievals combined together have also been used, providing data over the oceans. The advantage of combining both modes have shown to yield a stronger constraint at regional scales on $CO_2$ fluxes (Miller and Michalak, 2020). In addition, biases existing between these two modes of retrievals have been reduced (O'Dell et al., 2018). In this study, we focused on the nadir and glint modes, so the target mode is not discussed further. In addition, we only focus on the land nadir (LN) and land glint (LG) modes and do not use the ocean observation mode. Even if, since version 7, ocean biases in OCO-2 retrievals have been largely reduced (O'Dell et al., 2018), inversions assimilating OCO-2 ocean retrievals produced unrealistic results with annual global ocean sinks higher of $2.6 \pm 0.5$ GtC.yr$^{-1}$ compared to the state-of-the-art estimated in Le Quéré et al. (2018). Consequently, as for MIP v7 (Crowell et al., 2019), the OCO-2 ocean retrievals will not be further discussed in this study.

The algorithm developed to retrieve the column-average dry air mole fraction of $CO_2$ in the atmosphere (XCO$_2$) from the measured radiance spectrum comes from NASA's Atmospheric $CO_2$ Observations from Space (ACOS) project (O'Dell et al., 2012; Connor et al., 2008). The ACOS algorithm was first applied to GOSAT NIR and SWIR spectral measurements, which have similar spectral characteristics to the OCO-2 measurements, before being used for OCO-2 and OCO-3 (which were launched at later dates). In addition to the spectral data, ACOS uses meteorology and model data to constrain retrievals of XCO$_2$ along with a variety of other parameters such as aerosol optical depth, surface albedo, surface pressure and total column water vapor. In this paper, the modelers have used the ACOS bias-corrected retrievals (OCO-2 Level 2 Lite XCO$_2$ product) version 9 (Kiel et al. (2019); O'Dell et al. (2018), https://daac.gsfc.nasa.gov). Since October 2019, OCO-2 processing has used the ACOS version 9 (or "ACOS B9") algorithm, an update to the previous v7 and v8 versions (O'Dell et al. (2018)). Several changes have been applied in the v9 compared to the v7. In particular, the v8 data included corrections related the spectroscopy,





aerosol treatment, prior meteorology and the surface model. More details can be found in O'Dell et al. (2018). v9 included an

addition correction for the surface pressure estimation (Kiel et al. (2019)) which significantly reduced biases, particularly over

areas of rough topography. This bias correction in v9 allows a more uniformly distribution of $XCO_2$ over regions of interest,

decreasing the standard deviation to 0.74 ppm compared to v8 which was of 1.35 ppm.

     MIP modelers used all valid cloud-free OCO-2 retrievals (those considered as "good" by the quality_flag *xco2_quality_flag*=0)

from the OCO-2 Lite files and then selected the bias-corrected data (Wunch et al., 2011). Since the spatial resolution of OCO-2

data is much higher than the model grid box scale used in the inversions, the OCO-2 data are averaged to a coarser scale (in

this study, across a 10 second span, equivalent to about 67.5 km, along-track) before being assimilated. The retrieved column

$CO_2$, averaging kernels, prior $CO_2$ profiles, and a subset of the auxiliary parameters from the Lite files have all been averaged

across these 10-second spans in the same way, weighted by the inverse of the square of the retrieval uncertainty (variable

*xco2_uncertainty*) for each scene in the average. This is similar to the averaging done for the OCO-2 v7 MIP (Crowell et al.

(2019)), except that the 10-second averages are calculated directly, without the intermediate step of computing 1-second aver-

ages, as was done before. In computing the uncertainty to be placed upon the new 10-second-average $XCO_2$ value, an attempt

was made to account for correlations between the model-data mismatch (MDM) errors for each individual scene: each scene

within the 10-second span was assumed to have errors that were correlated with every other scene in the span with the same

positive correlation coefficient (+0.3 and +0.6 for scenes over land and ocean, respectively). Details of the form and deriva-

tion of these average uncertainties may be found in the 'constant correlation' section of Baker et al. (2021). The approach to

handling the correlations, while crude, represents an increase in complexity compared to what was assumed in the v7 MIP (no

reduction in uncertainty due to the averaging process, as described in Crowell et al. (2019)). Since it is known that the uncer-

tainty computed by the retrieval (in variable *xco2_uncertainty*) underestimates the true level of error in the retrieved $XCO_2$,

an additional term is added onto this "theoretical" uncertainty, in quadrature, to obtain a more realistic uncertainty per scene:

the standard deviation of all the $XCO_2$ values used in the 10-second average. In this *ad hoc* approach, scenes that have a very

small spread in $XCO_2$ values across the 10-second span are assigned the theoretical uncertainty from the retrieval, while those

for which the actual variability of the $XCO_2$ values is larger than the theoretical values are assigned a value closer to this

computed error level. Both resulting uncertainty is then passed through the formula to account for error correlations. Finally,

an additional term is added in quadrature to account for transport model errors. This model error term is computed from the

difference between the $CO_2$ concentrations computed by the TM5 and GEOS-Chem models when both are driven by the same

realistic surface $CO_2$ fluxes, after the annual mean difference filed is subtracted off; the values that result are considerably

smaller than those model errors added on for the OCO-2 v7 MIP Crowell et al. (2019). In contrast to this level of detail, the

uncertainties between different 10 s averages are assumed to be independent when assimilated into the inversions (Worden

et al., 2017; Crowell et al., 2019). Several studies have used this method in order to be coherent with the resolution of their

inversions or simulations regarding the OCO-2 resolution (Basu et al., 2018; Chevallier et al., 2019).





### 2.3 *In situ* CO$_2$ measurements

The set of *in situ* CO$_2$ measurements used for assimilation and for evaluation is drawn from 5 collections in ObsPack (Masarie et al., 2014, and https://www.esrl.noaa.gov/gmd/ccgg/obspack/) format. These component ObsPacks are:

1. **obspack_co2_1_GLOBALVIEWplus_v5.0_2019-08-12** (Cooperative Global Atmospheric Data Integration Project, 2019). This is the main source of *in situ* CO$_2$ measurements for MIP experiments, contributing 93% of all *in situ* measurements. It extends through the end of 2018.

2. **obspack_co2_1_NRT_v5.0_2019-08-13** (NOAA Carbon Cycle Group ObsPack Team, 2019): This near-real time ObsPack distribution are intended to provide data records after the end of the GLOBALVIEW+ v5.0 product, for laboratories and datasets that can provide measurement data outside of an annual update cycle. These data are provisional and generally have not undergone final quality control.

3. **obspack_co2_1_AirCore_v2.0_2018-11-13**. The balloon-borne AirCore instrument samples almost the entire atmospheric column. This early release collected all available profiles between 2014 and 2018.

4. **obspack_co2_1_INPE_RESTRICTED_v2.0_2018-11-13** (NOAA Carbon Cycle Group ObsPack Team, 2018) Aircraft
profiles at five sites in Brazil.

5. **obspack_co2_1_NIES_Shipboard_v2.1_2019-06-12** Continuous CO$_2$ analyzer measurements from 9 volunteer ships of opportunity operated by the Japanese National Institute for Environmental Studies (Tohjima et al., 2005; Nara et al., 2017).

.

This collection runs from January 1, 2000 to July 31, 2019 with an average of about 520 assimilable observations and 17 withheld observations per day (see Fig. 1.b). Measurements are contributed by 56 laboratories around the world. Measurements are collected at surface flask sites, at observatories and towers with continuous analyzers, onboard research and commercial ships, from light aircraft at regular profiling sites, and on commercial aircraft (see Fig. 1.a)

Only a small subset of ObsPack measurements are designated as suitable for assimilation, and the remainder as designated
for evaluation. The assimilable measurements meet two criteria: they can be successfully simulated in coarse-resolution global models, and they can be assigned a MDM error value. Many factors can render observations difficult to simulate in the global models used for this exercise. Sites located in areas with complex topography, or close to strong local sources like cities, or strongly influenced by small-scale circulation features such as land/sea breezes are all considered difficult to simulate. The CarbonTracker "adaptive model-data mismatch" scheme from CT2017 (Peters et al., 2007, with updates documented at
http://carbontracker.noaa.gov) was used to assign MDMs for this experiment. The MDM represents the expected statistical model residual from a measured value, and the current scheme estimates MDM values that vary by geographic location, month of year, local solar time of day, and distance from the earth surface. These values are developed using model performance from previous simulations, by computing a climatology of expected model errors for a given dataset. These model errors are





driven both by faults in simulated atmospheric transport and by incorrect upstream fluxes, but it is only the first of these that we attempt to represent with MDM. At many sites, model performance is dominated by a conspicuous cycle of seasonal error, attributed mostly to high ambient variability of $CO_2$ in the local growing season. Exploratory analysis has demonstrated that model-to-model differences in performance are significantly smaller than the other sources of variability in MDM like this annual cycle of model error.

The adaptive MDM scheme requires sufficient repeated measurements to develop a climatology of model performance, and as a result does not provide estimates of MDM error for measurements from temporary field deployments and aircraft campaigns (e.g. ACT-America (DiGangi et al., 2018) and ATom (Wofsy and ATom Science Team, 2018)). Many of these measurements without an MDM value are otherwise assimilable, since they sample background conditions that models should be able to simulate successfully. These data are particularly useful for model evaluation because they are generally independent of assimilated measurements.

## 2.4 Withheld data

In order to evaluate inversion model performance, a small subset of the assimilable *in situ* measurements were withheld for cross-validation. Each withheld measurement has an MDM value, which allows model residuals to be normalized by expected performance. The collection of withheld measurements was then used to evaluate the MIP ensemble.

Approximately 5% of the assimilable data were chosen for withholding. These were chosen carefully to maximize independence from the data designated for assimilation in the IS experiment. The criteria for withholding vary by measurement type. Flasks, which are generally taken on a weekly basis and with sampling criteria that emphasize background conditions, are already assumed to be independent from one another. All the measurements in a given aircraft profile are assumed to be related, so entire profiles were excluded. Quasi-continuous measurements, like those at towers and observatories, are assumed to be correlated in time, so all measurements during 5% of entire days were excluded.

Figure 2 shows the number of withheld data available for evaluation by latitude (Fig. 2, left panel) and by MIP region (Fig. 2, right panel). There are only about 1000 data points in the Southern hemisphere and approximately 600 in the tropics, in contrast to 5300 in the Northern hemisphere. For example, Fig. 2, right panel, shows the dearth of withheld observations for Tropical regions such as north and south Africa. There are no withheld data at all for some MIP regions, such as Northern Tropical Asia and Tropical South America.

## 2.5 ATom measurements

Atmospheric concentrations of $CO_2$ collected during the airborne campaigns of NASA's Atmospheric Tomography (ATom) mission (Stephens, 2017; Wofsy and ATom Science Team, 2018) are particularly useful for evaluation. These measurements come from four campaigns conducted over the Pacific and Atlantic oceans between 2016 and 2018. The ATom samples have been binned into five altitude levels (approximately of 0-1 km, 1-3 km, 3-7 km, 7-10 km and 10-14 km) and 9 latitude bins (every 15 degrees latitude) for evaluation (section 3.4). The density of ATom measurements by latitude and altitude bin are shown in Fig. 15 which will be discussed in section 3.4.

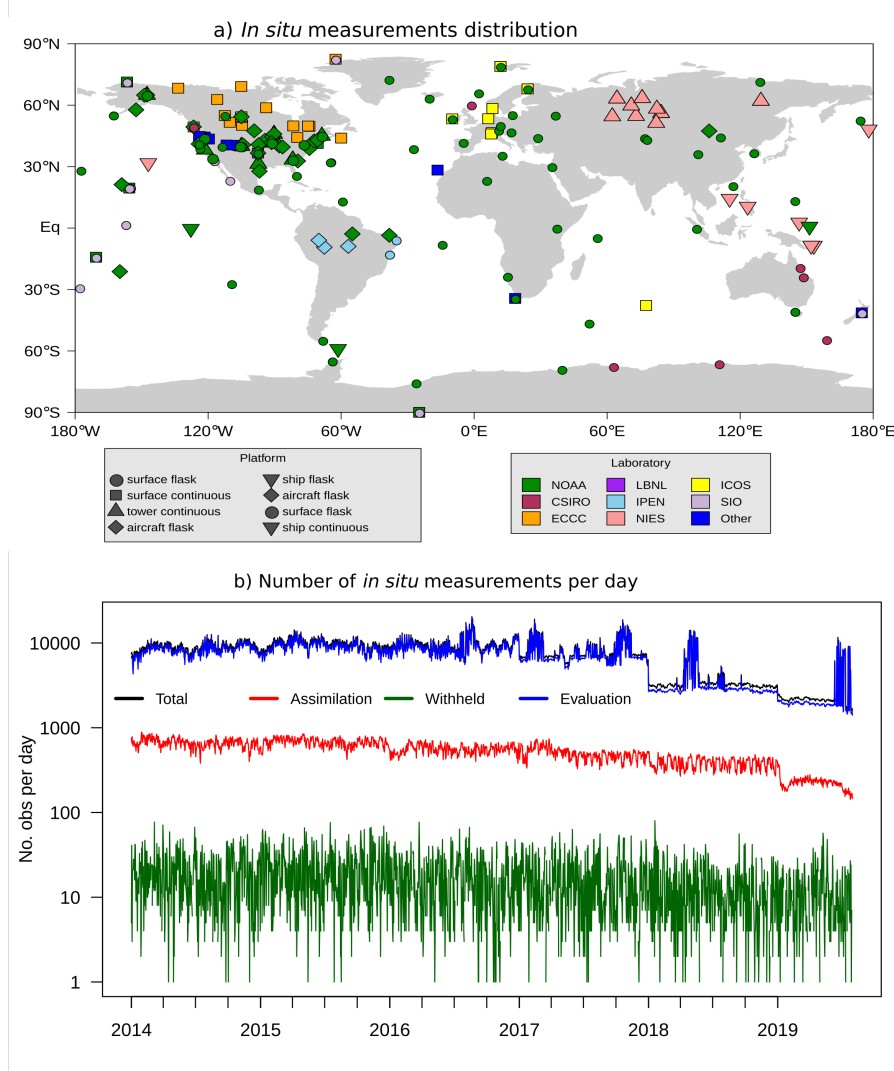

**Figure 1.** a) Distribution of assimilated *in situ* measurements around the world. The instrument platform is indicated by marker shape, whereas the color represents the laboratory collecting the data. NOAA is the United States National Oceanic and Atmospheric Administration, CSIRO is the Australian Commonwealth Scientific and Industrial Research Organisation, ECCC is Environment and Climate Change Canada, LBNL is the Lawrence Berkeley National Laboratory, IPEN is the Brazlian Instituto de Pesquisas Energeticas e Nucleares, NIES in the Japanese National Institute for Environmental Studies, ICOS is the European Union Integrated Carbon Observation System, and SIO is the Scripps Institute of Oceanography. Mobile shipboard programs are shown with a single marker at the mean location of the measurements. Figure from Jacobson et al. (2020a). b) Number of *in situ* measurements available per day, broken down by usage category. The total number of measurements (black) is the sum of assimilated (red), withheld (green), and evaluation (blue) data. The reduction in evaluation data at the end of 2018 corresponds to the end of available CONTRAIL measurements. The reduction in assimilated measurements at the end of 2019 corresponds to the transition from GLOBALVIEW+ to near-real time (NRT) data. Intermittent spikes in evaluation data are linked to campaigns like AToll and ACT-America.



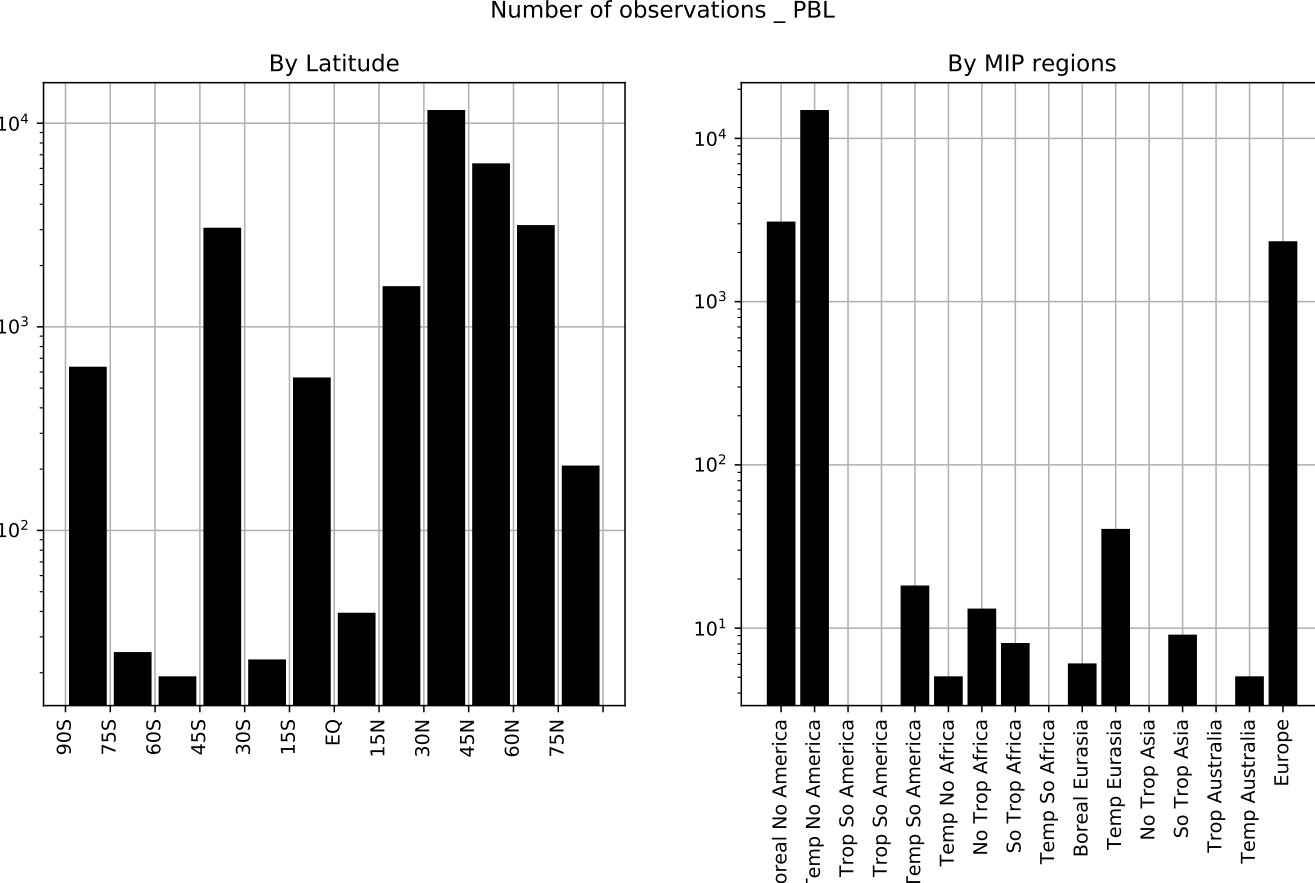

**Figure 2.** Number of withheld observations by latitude (left) and by MIP region (right) in the planetary boundary layer (PBL).

## 2.6 TCCON

The Total Carbon Column Observing Network (TCCON) is composed of around 30 sites around the globe estimating column-averaged dry air mole fraction of several atmospheric gases using ground-based remote sensing by Fourier Transform Spec-
trometers (Wunch et al., 2011). TCCON measures spectra of direct sunlight in the near infrared region. TCCON $CO_2$ retrievals are estimated to have precisions better than 0.25% (1-sigma) (Wunch et al., 2011). These retrievals have been used as the primary validation resource for several satellite missions, including OCO-2, SCIAMACHY and GOSAT (Wunch et al., 2011, 2017). OCO-2 observations were extensively evaluated against TCCON data in Wunch et al. (2017).

Posterior and prior concentrations are sampled for each 30-minute average TCCON retrieval before calculating the statistics
following the approach described in Crowell et al. (2019). For LNLGv9 inversions, the available 10s OCO-2 retrievals were averaged and compared to TCCON observations with a 5° latitude and longitude geometric coincidence criterion and within





**Table 3.** Geolocation and reference of each TCCON station used for the evaluation section.

| TCCON sites | Country | Latitude | Longitude | Reference |
|---|---|---|---|---|
| Eureka | Canada | 80.05N | 86.42W | Strong et al. (2014, 2017) |
| Ny-Ålesund | Spitsbergen | 78.9N | 11.9E | Notholt et al. (2017) |
| Sodankylä | Finland | 67.4N | 26.6E | Kivi et al. (2014) |
| Białystok | Poland | 53.2N | 23.0E | Deutscher et al. (2014a, b) |
| Bremen | Germany | 53.10N | 8.85E | Notholt et al. (2014) |
| Karlsruhe | Germany | 49.1N | 8.4E | Hase et al. (2014a, b) |
| Paris | France | 48.8N | 2.4E | Te et al. (2014) |
| Orléans | France | 47.9N | 2.1E | Warneke et al. (2014) |
| Garmisch | Germany | 47.5N | 11.1E | Sussmann and Rettinger (2014) |
| Park Falls | Wisconsin (USA) | 45.9N | 90.3W | Wennberg et al. (2014a) |
| Rikubetsu | Japan | 43.5N | 143.8E | Morino et al. (2017, 2014c, d) |
| Lamont | Oklahoma (USA) | 36.6N | 97.5W | Wennberg et al. (2016, 2014d) |
| Anmeyondo | Korea | 36.5N | 126.3E | Goo et al. (2014) |
| Tsukuba | Japan | 36.1N | 140.1E | Morino et al. (2014b, a) |
| Edwards | California (USA) | 34.2N | 118.2W | Iraci et al. (2016, 2014) |
| Caltech | California (USA) | 34.1N | 118.1W | Wennberg et al. (2014b, c) |
| Saga | Japan | 33.2N | 130.3E | Kawakami et al. (2014) |
| Izaña | Tenerife | 28.3N | 16.5W | Blumenstock et al. (2014) |
| Ascension Island | | 7.9S | 14.3W | Feist et al. (2014) |
| Darwin | Australia | 12.4S | 130.9E | Griffith et al. (2014a) |
| Réunion Island | | 20.9S | 55.5E | De Mazière et al. (2014) |
| Wollongong | Australia | 34.4S | 150.9E | Griffith et al. (2014b) |
| Lauder 1202HR | New Zealand | 45.0S | 169.7E | Sherlock et al. (2014) |

1h of the overpass. All TCCON sites used in the evaluation section are listed in Table 3. Figure 3 illustrates the different time ranges and observation numbers across TCCON sites during the 2015-2018 period.

## 3 Results

We discuss results from two v9 MIP experiments: IS, in which only the *in situ* $CO_2$ measurements are assimilated, and LNLGv9, for which OCO-2 v9 land nadir and land glint retrievals were assimilated together. The v9 MIP simulations are conducted over the four years from 2015-2018. For comparison, we also include results from the v7 MIP LNv7, and LGv7 experiments (Crowell et al., 2019), although those results are limited to 2015 and 2016 only. In both the v7 and v9 MIPs, ocean glint retrievals were also assimilated in separate experiments. Those experiments will not be discussed here, as the ocean





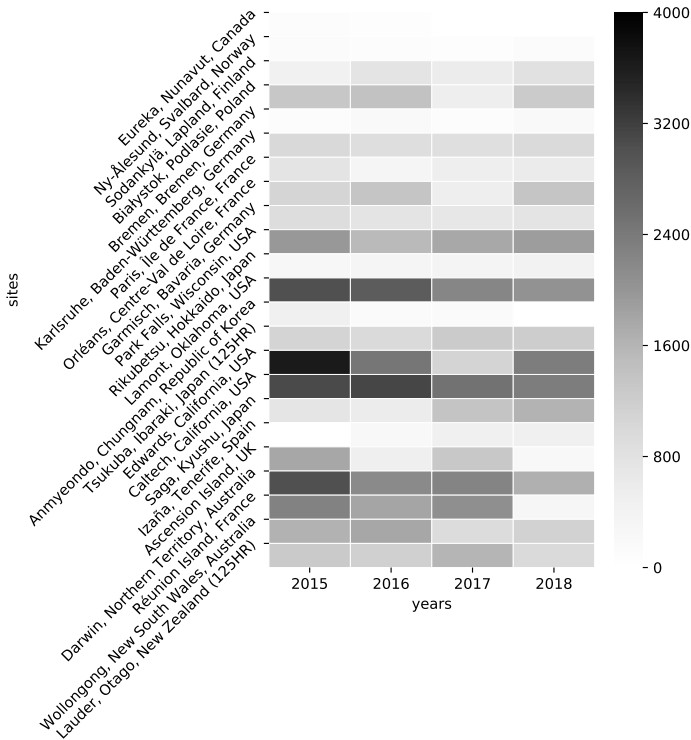

**Figure 3.** Number of TCCON observations for each site and year.

glint retrievals have uncharacterized biases in both v7 and v9 versions. For the purpose of analysis, the standardized fossil fuel emissions have also been removed from all prior and posterior fluxes. This section is organized as follows: we first analyze fluxes at the global scale before moving to three broad zonal bands. We then finish with a regional flux analysis. In order to evaluate the spatio-temporal variability of regional fluxes, the different modeling groups' flux estimates have been aggregated from their individual model grid boxes up to OCO-2 MIP regions (see Fig. 4) similar to those used in the MIP v7 analysis from

Crowell et al. (2019).

### 3.1 Global flux estimates

Figure 5 represents the annual emissions (in PgC/yr, for the left panels) and the monthly emissions (in PgC/month for the right panels) at the global scale for each experiment. As expected, at the global scale the posterior fluxes of all OCO-2 observation types, as well as the prior, and IS emissions, have similar seasonal cycle (Fig. 5.b). Fluxes for all of the models are well

constrained at this scale as they are the difference of the relatively well know fossil fuel fluxes and the well-measured global atmospheric increase. However, the peak sinks during the Northern Hemisphere growing season (from May through September) are slightly larger with OCO-2 v7 than with OCO-2 v9. Additionally, while the growing season observed with v7 is shifted earlier in the year relative to IS and the unoptimized prior fluxes, this is not the case with v9 (although the priors in the v7 and





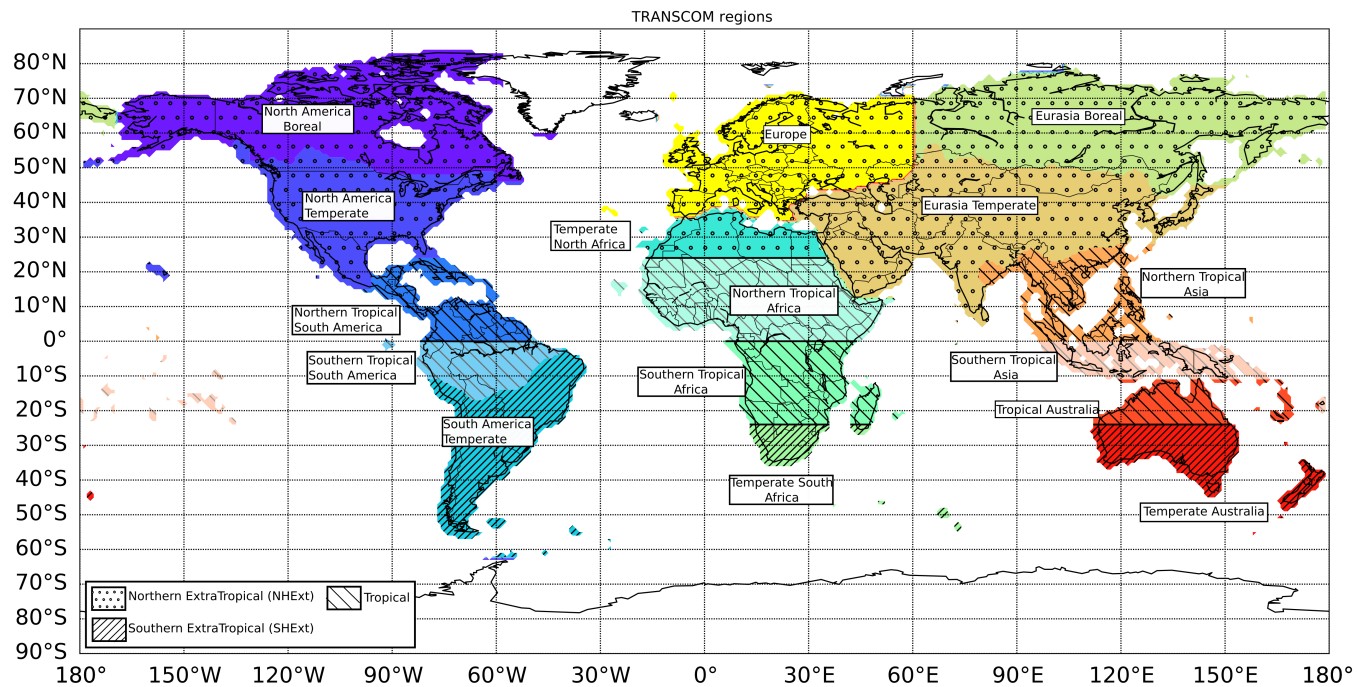

**Figure 4.** OCO-2 MIP regions to which gridded fluxes are aggregated for comparison and evaluation.

v9 MIPs are not the same for some models, this could also be due to the longer number of years in the v9 experiments and the

models participating in the two MIP versions). While the median values for the prior emissions of the land plus ocean fluxes in v7 were around -2.5 PgC/yr in 2015-2016 (Fig. 3.a in Crowell et al. (2019)), they are around -3.75 PgC/yr for the same period (2015-2016) in v9 (Fig. 5.a) and have a smaller ensemble spread among the models.

For the annual total fluxes, at the global scale, we can observe a good agreement between IS and all OCO-2 fluxes in 2015 with emissions of approximately -3.75 PgC/yr. However, in 2016, LNLGv9 gives a stronger sinks of -4.5±0.1 PgC/yr compared

to IS and v7 with sinks median around -4 PgC/yr. This stronger sink observed with v9 in 2016 comes from Southern Extratropics (Fig. 6.e). For 2017 and 2018, both IS and v9 are very close to each other with total annual sinks between -5.5±0.1 and -4.5±0.1 PgC/yr. The ensemble spread among the models for the annual fluxes is smaller with v9 than with v7, which could either be due to the longer span of data inverted in v9 where the ability to compute the trend (or the total land+ocean flux) improves, reducing noise in the estimates, or suggest better agreement between the models for the v9 version. With a

4-year record of flux estimates from both IS and LNLGv9, we are able to group 2015 and 2016 together. These are years which have been associated with a large $CO_2$ growth rate due to a strong El Niño event (Malhi et al., 2018) compared to 2017 and 2018. For the rest of this study, we will call the 2015-2016 period the "El Niño period" and the 2017-2018 period the "recovery period". Several previous papers already studied these periods, focusing on the impact of El Niño over the tropics (Liu et al., 2017; Palmer et al., 2019; Wigneron et al., 2020). The contrast between these two different periods can also be observed over



the Global Land in particular, where the difference between the El Niño period and recovery is particularly large in the IS
inversions. Global Land (Fig. 5c and d) and Global Ocean (Fig. 5e and f) show compensating effect, where v9 finds higher
sources during the El Niño period over the land that are balanced with stronger sinks over the ocean. Generally v9 yields a
weaker global land sink and stronger ocean sink. During the recovery period, IS gives a stronger median sink (3.5 PgC/yr) than
v9 (1.75 PgC/yr). Friedlingstein et al. (2019) estimated a global land uptake of around 2 PgC/yr in 2018 over the global land

for the net fire and biospheric fluxes, which is closer to what we see constrained by the v9 data.

## 3.2 Latitudinal bands

As mentioned in Crowell et al. (2019), the observations should constrain the fluxes over latitudinal bands more effectively than
longitudinally or by geopolitical region, due mainly to the effects of prevailing zonal winds across much of the globe. OCO-2
observations are collected across the sunlit portion of the Earth crossing all zonal bands about 15 times per day. Hence we split

our analysis between the tropics and extra-tropics, examining only land emissions since we expect the data constraint to be
strongest there for the mostly land-based observations that we discuss. Over the Northern Extra-Tropics (Fig. 6b), we observe
flux dynamics similar to those on the Global scale, with large seasonal variation and deeper sinks during summer for OCO-2
and IS inversions compared to the prior. All experiments put the land sink more in the Northern extra-tropics than in the tropics
and Southern extra-tropics (Tans et al., 1990).

For the annual fluxes (Fig. 6a), we can see that IS and LNLG v9 Northern Extra-Tropics fluxes are close to each other in
2015-2016, with an increase of net carbon uptake relative to the prior of around $2.5 \pm 0.25$ PgC/yr to $3 \pm 0.25$ PgC/yr in 2015
and 2016, respectively. In 2017 and 2018, we can observe a decrease of net carbon uptake larger with v9 (fluxes of around 2
PgC/yr in 2017 and $1.75 \pm 0.25$ PgC/yr in 2018) than IS (2.75 PgC/yr in 2017 and $2.5 \pm 0.25$ PgC/yr in 2018).

We also see some differences between v7 and v9. LN v7 fluxes are closer to what is observed in both LNLG v9 and IS during

the El Niño period, with a median sink around -2.25 PgC/yr, but LGv7 gives a weaker sink (-1.5 PgC/yr) due to larger sources
in fall 2015 and 2016. As we can observe for all other latitudes bands and we will observe for smaller regions, LNLGv9 tends
to be closer to LNv7 than to LGv7. This points to previously known issues with the v7 LG data that were resolved with a
unified bias correction in OCO-2 v9. Interestingly, the seasonality for the v9 results more closely aligns with the IS results, and
the large efflux at the end of the growing season in v7 LNLG has disappeared in v9.

The Southern extra-tropics (SHExt, Fig. 6e and f) are known to have fewer IS observations as well as little land mass
(Crowell et al., 2019), and hence fewer land retrievals to constrain the fluxes, which are signifcaintly weaker over this latitude
band. This could explain why, for this latitude, the prior, IS, v7 and v9 results have different seasonality. LNv7 and LNLGv9
have different seasonality and v7 has a delay in the efflux peak for 2015 and no efflux peak at all for 2016. The seasonal
amplitude observed with LNLGv9 is smaller for the whole period compared to IS and LGv7 and has a delay in 2017 and 2018

compared to IS. The differences in monthly emissions are also observed in the annual fluxes. They show, for the whole period,
stronger sinks with v9 than with IS, and v7. However, in contrast to NHExt, the ensemble spread is larger with v9 than with v7.
The bias reduction of v9 gives a smaller spread and hence a better agreement among the models, particularly over the Northern
Hemisphere.





**Figure 5.** Monthly median fluxes in PgC/month (right side) and annual flux in PgC/yr (left side) from 2015 through 2018 for the different experiments : Prior (black) and posterior ensemble fluxes constrained by *in situ* data (red), OCO-2 v7 LN (cyan with hatched cross), OCO-2 v7 LG (green with hatched), and OCO-2 v9 LNLG (blue) retrievals. LoFI fluxes (purple circles) are plotted alongside IS fluxes. For both the monthly time series and the annual fluxes plots, the shaded bar represents the range of emissions among the models and the solid lines represent the median of the model ensemble for both annual and monthly plots. Top plots are for global (Land + Ocean), middle plots for Global Land and bottom plots for Global Ocean.





Over the tropics (Fig. 6c and d), the seasonal peak efflux is typically in the fall, with an anomalously strong source in fall
2015 during the El Niño intense period. On average, the seasonality seems to be similar for the IS and OCO-2 inversions, but
different from that of the prior. However, for the whole period, the v9 OCO-2 annual mean source is about 0.5±0.1 PgC/yr
stronger than for IS. OCO-2 observations have a more frequent coverage over the tropics than the *in situ* network. However,
OCO-2 retrievals can be biased due to cloud coverage during the wet season and aerosol from biomass burning during the dry
season (Merrelli et al., 2015; Massie et al., 2017). LNLGv9 gives stronger annual sources, particularly for the El Niño period,
and with a smaller ensemble spread, than does v7. The OCO-2 LGv7 ensemble spread does not deviate from the prior spread,
showing the large impact of v9 corrections on the retrievals. The *in situ* Obspack data set used for this study has been updated
and include more data per site, contrary to the *in situ* data set used in Crowell et al. (2019) for the v7 MIP. We then have
stronger sources observed with ISv9 than with ISv7 in 2015 and 2016. In addition, we can observe with both IS and LNLGv9
two clearly-distinguished periods in terms of annual mean flux: the El Niño period for 2015 and 2016, with sources between
1.5 PgC/yr and 2 PgC/yr, and the recovery period, with median values between -0.5 and 0.5 PgC/yr. For both monthly and
annual fluxes, large sources of carbon are observed over the tropics for the whole period of study. Wigneron et al. (2020) found
that the pan-Tropical above-ground carbon stocks in the tropical humid forests did not recover after the 2015-2016 El Niño,
due presumably to a combination of deforestation and climate conditions.

## 3.3 Fluxes by region

### 330 3.3.1 Northern extra-Tropical region

We see similarities in the seasonality and annual flux across zonal bands in the OCO-2 (mainly v9) and IS results for the
Northern extra-tropics, but also differences between v7 and v9: now we look at smaller spatial scales to see where these
agreements or disagreement are observed. Figure 7 shows monthly and annual fluxes for Northern America (top panel), Europe
(middle panel) and Northern Asia (bottom panel).

Over Northern America (Fig. 7a and b), monthly and annual fluxes show different patterns for all data types. Prior annual
median fluxes are around -0.25 PgC/yr for 2015-2018. Median values for LNLGv9 show 0.5 PgC/yr stronger sinks during the
El Niño period, with LNv7 showing even deeper sinks. IS and LGv7 have deeper sinks than the prior but weaker than LNv7
and v9.

Over Europe, we can see that IS agrees better with LNLGv9, with similar annual median fluxes. For this region, v7 and
v9 are particularly different, as v9 gives larger sinks during summertime. Interestingly, LNLGv9 seems then to be in a better
agreement with what was observed in Houweling et al. (2015) than v7. Houweling et al. (2015) assimilated GOSAT data over
the 2009 and 2010 period and observed a larger carbon uptake for Europe with GOSAT than with in-situ data, as was also
observed by Chevallier et al. (2014) and Reuter et al. (2014). Inferred fluxes using v9 seem then to be more consistent with
other studies, but more analysis is needed to understand why this difference between v7 and v9 appears over Europe (which
could be due to a dipole between Europe and Northern Africa as observed and mentioned by the previous studies of Houweling



**Figure 6.** Same as Fig. 5 but for the zonal band : Northern Extra-tropics (NHExt, top panel) with latitudes from 23°N-90°N, Tropics (middle panels) with latitude from 23°S-23°N, and Southern Extra-tropics (SHExt, bottom panels) with latitude from 90°S-23°S.





et al. (2015); Chevallier et al. (2014); Feng et al. (2016); Reuter et al. (2014, 2017)) but not for Northern America and Northern Asia.

For Northern Asia (including Eurasia Temperate and Eurasia Boreal), while IS gives large sinks (with an ensemble spread between -2.5 PgC/yr and -0.5 PgC/yr for the whole period), v9 and v7 both show weaker sinks (with a ensemble spread between -1.25 PgC/yr and -0.25 PgC/yr for 2015 and 2016) and a decrease with v9 for 2017 (-0.5 +- 0.5 PgC/yr) and 2018 (-0.25 +- 0.5 PgC/yr). The 2017 and 2018 LNLGv9 emissions are closer to the priors. For the El Niño season (2015 and 2016), LNv7 has the same annual emissions as LNLGv9 but with a smaller ensemble spread; however, LGv7 shows weaker sinks with particularly strong emissions during the Fall, which could be due to either fewer observations or a possible bias at higher latitudes during the Northern hemisphere winter. The disagreement between the OCO-2 and *in situ* inversions might be driven by the differences in the amount of data assimilated, since both inversions have same transport model and inverse set-up. We know that there are fewer *in situ* than OCO-2 observations above Northern Asia, and particularly above the boreal forest of Eurasia, which is an important area for sources and sinks of atmospheric $CO_2$ (Houghton et al., 2007; Siewert et al., 2015). The combination of sparse data, in an area not well observed, with transport uncertainty could be the cause. We see that the ensemble spread of the models is larger for IS than for v7 or v9 and that the annual fluxes differ by almost 1.5 PgC/yr between the IS and OCO-2 inversions during the El Niño period. This is not the case for Europe and Northern America, where the surface measurements are most densely concentrated (Fig. 1.a). Finally, due to the additional two years of fluxes using IS and v9, we see a trend towards a weaker sink from 2015 to 2018 for Northern Asia that is not observed for Europe or North America.

### 3.3.2 Tropical region

We now look at the Tropics split across the Northern and Southern hemispheres. The Southern and Northern tropics are represented in Fig. 8, with annual fluxes on the left side (Fig. 8a and c) and monthly fluxes (Fig. 8b and d) on the right side of the plots. As observed earlier for the Tropical band, and in contrast to what was found in Crowell et al. (2019) using the *in situ* data, the IS inversions give similar seasonal amplitude and annual mean emissions as OCO-2 for both the Northern and Southern Tropics in 2015 and 2016, the El Niño period. However, for 2017 and 2018, IS seems to follow the pattern of the prior at the annual timescale. This difference, particularly for the Northern Tropics, in 2017-2018 seems to suggest a different signal observed with IS compared to v9. We can also observe that the ensemble spread is almost the same between both version v7 and v9 for both tropical bands. In both tropical hemispheres, LNv7 gives relatively more net $CO_2$ emitted to the atmosphere than LGv7. LNLGv9 agrees with LNv7 over 2015-2016 for the Northern tropics, but gives more of a source (0.5 PgC/yr) than LNv7 (0.1 PgC/yr) and LGv7 (-0.4 PgC/yr) over the Southern Tropics. This 0.5 PgC/yr source of carbon observed with v9 is also observed with IS, suggesting that more carbon could have been released during the El Niño period than previously inferred with v7. During the recovery period in the Northern Tropics, v9 only has net sources of carbon while for IS some models have sinks of up to -1 PgC/yr. In contrast, over the Southern Tropics, the median $CO_2$ flux values given by IS are positive, while they are strongly negative for v9.

In order to see the resolution provided by the data at finer scales in the tropics, we examine fluxes for six tropical regions (three over the Northern Tropical hemisphere (Fig. 9) and three over the Southern Tropical hemisphere (Fig. 10)).





**Figure 7.** Same as Fig. 5 but for the Northern Extra-tropics regions : Northern America (top panel), Europe (middle panels), and Northern Asia (bottom panels).





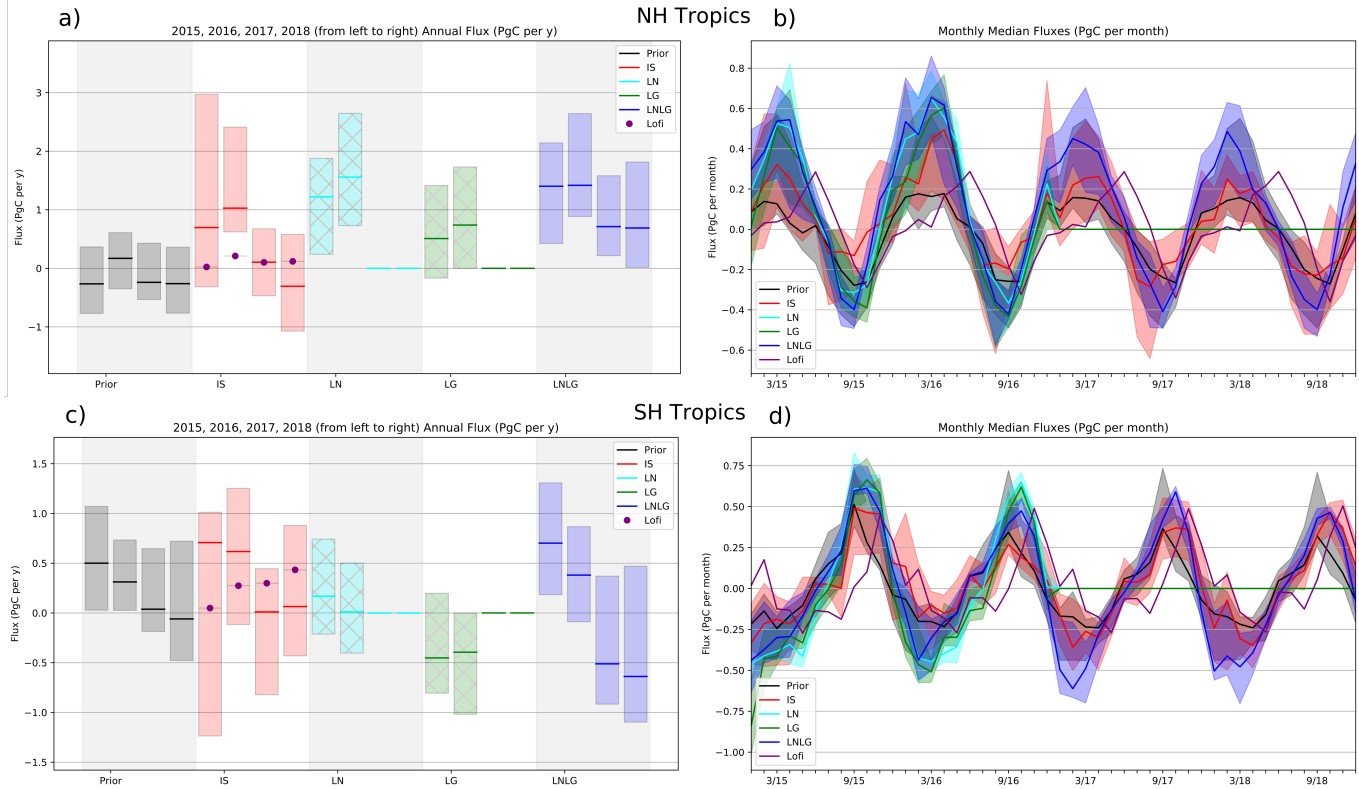

**Figure 8.** Same as Fig. 5 but for the Tropical band : Northern tropics (top panel) with latitudes from 0°N-23°N, and Southern tropics (bottom panels) with latitude from 23°S-0°S.

Figure 9 shows the monthly and annual fluxes over Northern Tropical South America, Northern Tropical Africa and Northern Tropical Asia. When we look at the annual fluxes of the Northern Tropical regions (Fig. 9a, c, e), we do not observe significant differences between v7 and v9 inversions with respect to the ensemble spread and the median values, except for Northern Tropical South America, where v9 has a slightly higher net $CO_2$ outgassing (0.5 PgC/yr) than v7 (around 0.4 PgC/yr for LNv7 and 0.2 PgC/yr for LGv7). Here again, we find better agreement between LNv7 and LNLGv9. The IS and OCO-2 annual fluxes have a similar temporal pattern over North Tropical South America, with average fluxes close to 0.5 PgC/yr and a smaller ensemble spread for OCO-2 than for IS. But the most obvious differences appear between the IS and OCO-2 inversions over Northern Tropical Africa and Northern Tropical Asia. The OCO-2 inversions give a larger source of carbon over Northern Tropical Africa compared to the IS inversions, similar to conclusions from the v7 MIP (Crowell et al., 2019). These large inferred emissions are consistent with Palmer et al. (2019); Wigneron et al. (2020), who found that the Africa continent accounted for 56% of carbon emissions during the 2015 El Niño event. However, for Northern Tropical Asia, both v7 and v9 give sinks of carbon (around -0.25 PgC/yr for LNv7 and v9 and -0.4 PgC/yr for LGv7), while IS gives a source of around 0.25 PgC/yr during the El Niño season. Northern Tropical Asia is the only region where we found a change in fluxes





between the v7 and v9 *in situ* inversions. Indeed, the v7 IS inversions Crowell et al. (2019) had a median values for 2015 and 2016 of -0.10 PgC/yr over Northern Tropical Asia. Sparse *in situ* coverage over the Tropical regions compared to the Northern hemisphere (Fig. 1.a) could explain this difference with the OCO-2 inversions, but further analysis over this region needs to be done. The monthly seasonality is more similar between all experiments for Northern Tropical Africa than it is for the two other Northern regions.

For the monthly emissions of the Southern Tropical regions (Fig. 10), we can see the strong impact of El Niño in fall 2015 over Southern Tropical Asia in the larger emissions (with a maximum of 0.35 ±0.01 PgC/yr) given by all the inversions compared to the rest of the period. This large emission in fall 2015 mainly come from Indonesian fires. Field et al. (2016) estimated fire emissions in 2015 over Indonesia to be 380 TgC. This El Niño impact started in the end of 2014, peaked in fall 2015, and ended in May 2016 (Liu et al., 2017). The impact of the El Niño is particularly noticeable with the long period available from v9 compared to the two years from v7. In addition, this peak is reflected in the annual mean fluxes, where v9 gives a strong separation between 2015 and the rest of the period (2016 and the recovery period), which is also observed with the IS. Annual median fluxes between v7 and v9 are almost similar, with sources in 2015-2016 of around 0.2 PgC/yr (-0.05 PgC/yr) with LNv7 (LGv7) and of 0.4 PgC/yr with v9. Similarly, even if there is a change in data coverage between the two versions, we do not observe much difference between v7 and v9 for Southern Tropical South America (Fig. 10a and b), except that the ensemble spread is larger with v9 than with v7. The bias correction in v9 allowed to have more data than v7 over the Amazon in order to pass the quality flag criteria (Miller et al., 2018; O'Dell et al., 2018). In addition, we can observe a different amplitude in the seasonality between IS and OCO-2 for this region. This difference over Tropical Southern America might be due to the cloud effect (Crowell et al., 2019) of the wet season over the Amazon affecting OCO-2 data. But this could be also because most of the *in situ* data are located mainly inside the Amazon and not in the Cerrado savanna of Brazil, resulting in IS inversions being dominated by tropical forest seasonality. Alternatively, the OCO-2 inversions could be dominated by savanna seasonality (Baker et al., 2021a *in prep*). Finally, over Southern Tropical Africa (Fig. 10c), we obtain a large difference between the annual means of v7 and v9 that we did not observe for the other regions. While LN and LG v7 give a sink of carbon during the El Niño period of around -0.25 PgC/yr, LNLGv9 gives a sources of around 0.25 PgC/yr, which seems not to be compensated by a flux signal in Northern Tropical Africa. These sources of carbon come from weaker sinks during the growing season (from November through March). This source of carbon during the El Niño period is also observable with the IS inversions (and was also observed with the ISv7, Crowell et al. (2019)).

## 3.4 Evaluation against independent data

To assess the accuracy of the posterior flux results presented previously, we evaluate them here by sampling the resultant posterior concentrations and comparing them to withheld data, ATom aircraft measurements and TCCON data. All modelers have sampled their posterior concentrations at the times and locations of the evaluation data.

### 3.4.1 Withheld *in situ* evaluation data

Here, we evaluate against the withheld *in situ* data introduced in Section 2.4.

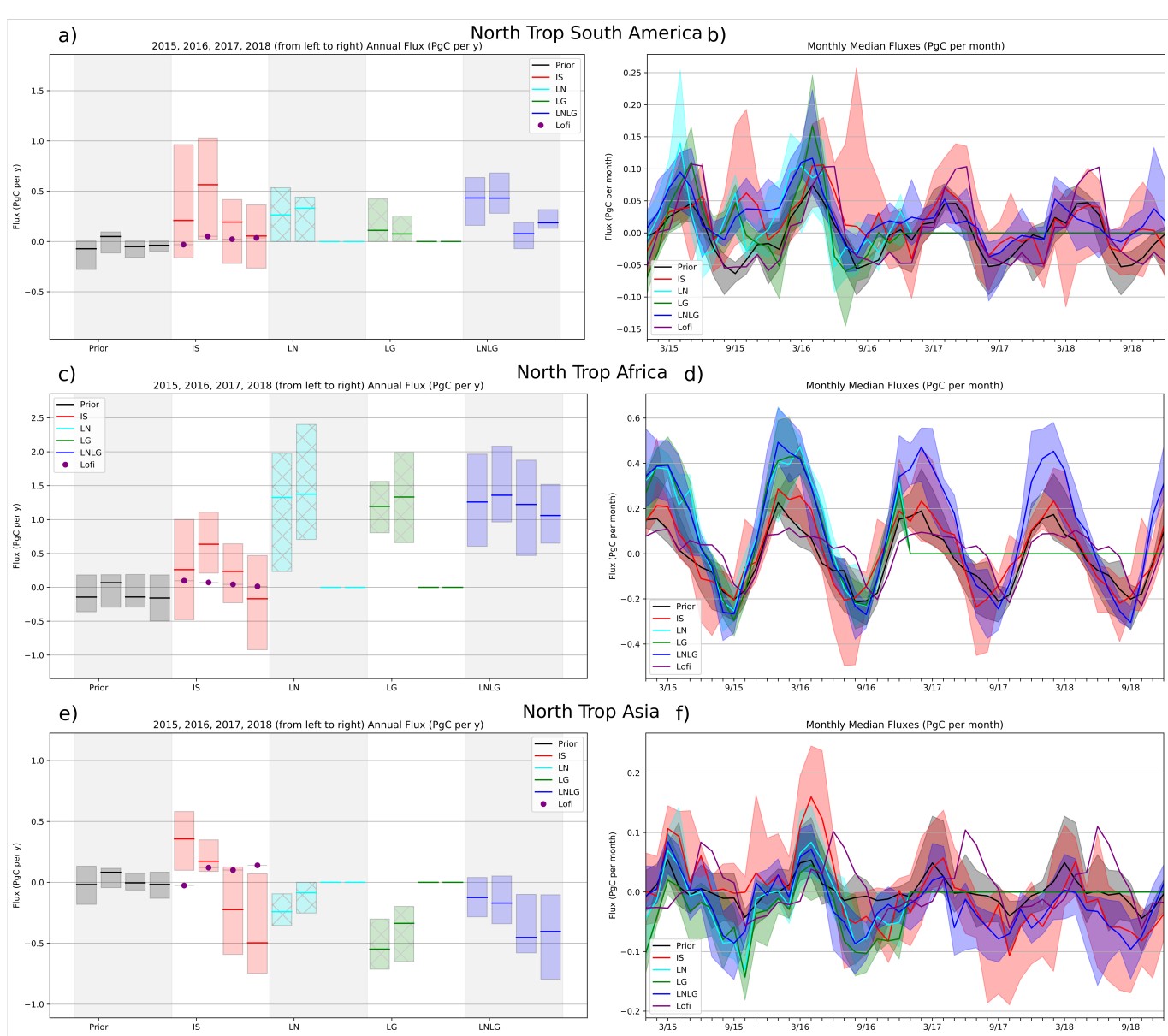

**Figure 9.** Same as Fig. 5 but for the Northern Tropical regions : Northern Tropical South America (top panel), Northern Tropical Africa (middle panels), and Northern Tropical Asia (bottom panels).

**Figure 10.** Same as Fig. 5 but for the Southern Tropical regions : Southern Tropical South America (top panel), Southern Tropical Africa (middle panels), and Southern Tropical Asia (bottom panels).





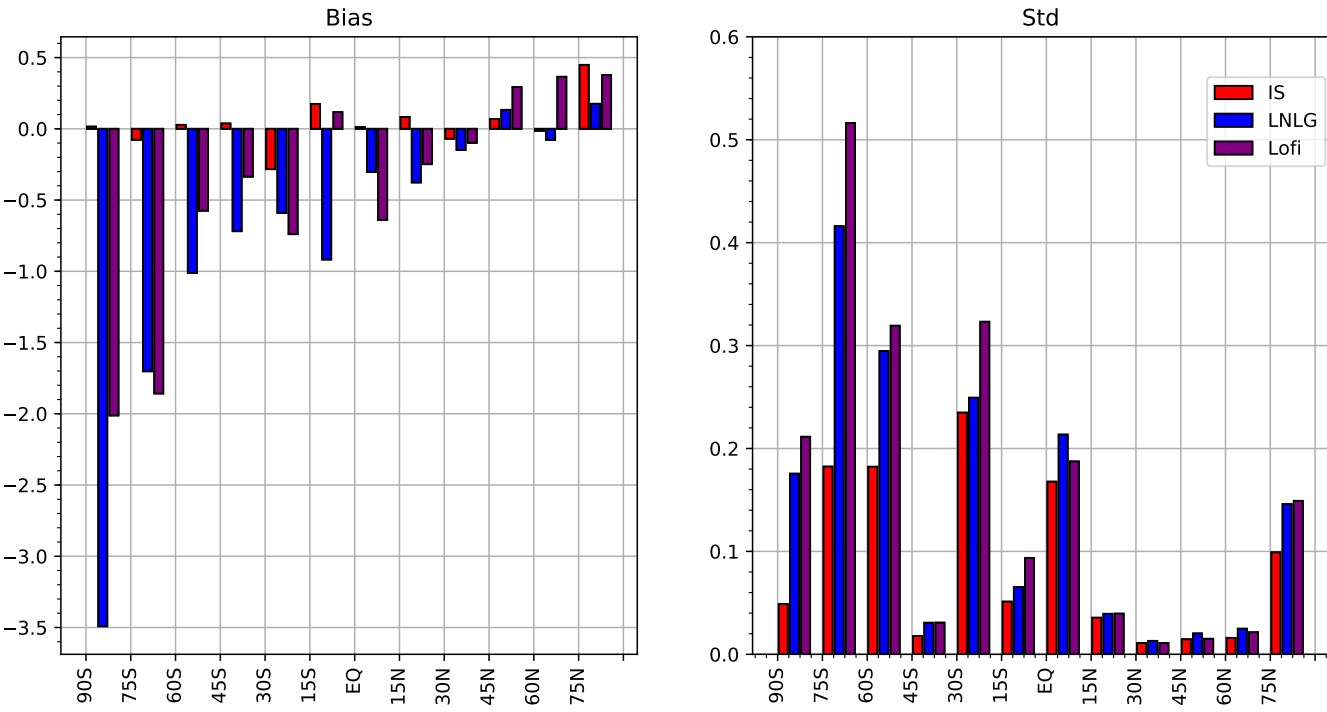

**Figure 11.** Normalized bias (model minus observation divided by model data mismatch (MDM, see Section. 2)) in the left panel and standard deviation in the right panel for the ensemble mean of IS (in red), LNLGv9 (in blue) and LoFI (in purple). The evaluation is done by latitude and in the PBL. The data have been averaged over the 4 years of study (2015-2018).

Figure 11 shows the evaluation with the withheld data by latitude band for IS, and LNLGv9. Over the southern hemisphere, a large underestimation of the ensemble mean of LNLGv9 appears compared to the observations. While biases are larger over the southern hemisphere they are smaller over the tropics. The large biases observed in LNLGv9 and not with the IS, could be due to a latitudinal bias in the OCO-2 data. Going from the southern to the northern hemisphere, we can see a change in the

v9 biases, which go from underestimation to overestimation by the models. This behavior also appears in the IS but with lower biases (less than 0.5 for all latitudes). The variability between the models and observations is higher in the southern hemisphere than in the tropics and the northern latitudes (Fig. 11, right plot). Overall, the IS experiments are more consistent with the *in situ* measurements than the LNLG experiments are.

Figure 12 shows the normalized bias and standard deviation of the ensemble mean by MIP regions for LNLGv9 and IS.

Evaluation by MIP regions reveals a different behavior than observed by latitudes. For instance, over Temperate North America, v9 seems to have a larger underestimation of the observations than IS has, while for Temperate South America and Northern Tropical Africa, v9 has larger underestimation than IS. Over Europe, we see here that v9 has smaller biases (less overestimation)



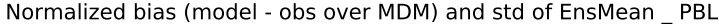

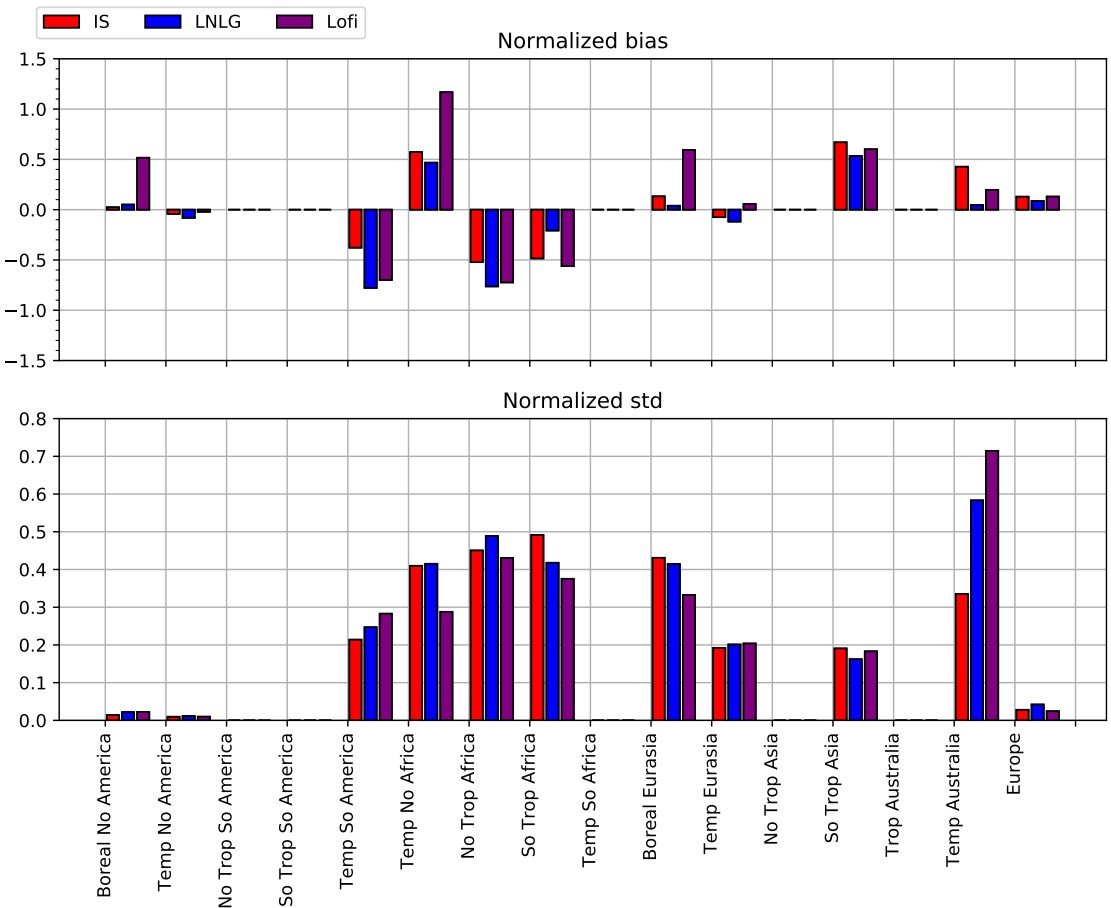

**Figure 12.** Same as Fig. 11 but instead of an evaluation by latitude, the evaluation is by MIP region.

than IS. Finally, if we look at Southern Tropical Africa, v9 is less biased than IS. Regarding the standard deviation, it seems that the three experiments have similarities, but with closer agreement between IS and v9 for some regions.

Since the MDM values range over two orders of magnitude, the use of the normalized residuals gives the most meaningful interpretation of the residuals. When we look at the normalized bias and standard deviation for each models between IS and LNLGv9 experiments by latitudes (Fig. 13 a, b, c, and d) and by MIP regions (Fig. 14 a, b, c, and d), we can see this large difference with underestimation for all models assimilating v9 in the southern and tropical latitudes. This underestimation with all models is, however, generally not obtained with IS. When comparing the root-mean-square error (RMSE) between

the models and the withheld data (Fig. 13 e,f), we can observe higher value for all models between 30°N and 75°N with RMSE between 6 ppm (4 ppm) and 10 ppm (7 ppm) for LNLG inversions (for IS inversions respectively), while RMSE are below 3ppm in the southern latitudes and in the tropics. This larger raw errors observed in the northern latitudes could come





from the difference in the *in situ* network between northern and southern latitudes. Indeed, there are more measurements in the northern hemisphere close to regionally significant sources and sinks compared to other latitudinal bands (tropics and

southern hemisphere). Compared to the ensemble mean, every model shows small normalized standard deviations across the latitudes, with values near 0.1 in the Northern Hemisphere and going from 0.2 to 0.6, according the models, between 75°S to 60°S. However, the evaluation by MIP regions shows similarities between IS and v9 for all regions. This similarity is also found for the RMSE values (Fig. 14 e,f) with however larger values for LNLG (maximum values of 10ppm) than with IS (maximum of 8ppm) over temperate north America, boreal Eurasia, boreal north America, Europe and southern tropical

Asia. Additionally, errors are larger for temperate north America than for temperate Eurasia as expected due to the sampling distribution. Normalizing the models with the MDM's values is equivalent to normalize against this expected variability. We can also observe that some models do better at some latitudes and regions, while the others are better at other regions. However, the goals of the study are to envelope the uncertainty in the inversions, rather than ranking model performance.

### 3.4.2 ATom evaluation

Figure 15 shows the comparison of the posterior concentrations that were sampled at the locations and times of the ATom flight campaigns during 2015-2018. IS inversions overestimate $CO_2$ concentrations for almost all latitudes and altitudes, with the exception of an underestimation between 30°N and 60°N at low altitudes. On the other hand, LNLGv9 shows an underestimation at almost every latitude and gives larger biases than IS in general. This underestimation is consistent with the large underestimation observed, particularly in Southern Hemisphere, with the withheld data (Fig. 13). In addition, for both experiments, we

observe a large overestimation (with biases of 1 to 2 ppm) in the stratosphere of the boreal latitudes. This stratospheric region is affected by atmospheric circulation and has few observations to constrain the inversions. It is possible then that this excess of concentration, in both experiments, reflects the initial conditions of the inversion.

### 3.4.3 TCCON evaluation

Figure 16 shows bias and standard deviation for the prior, IS and LNLGv9 experiments over all TCCON sites available during

the four years of the study. We can observe that prior concentrations are biased high for almost all TCCON sites and all models. Compared to the evaluation of ISv7 in the study of Crowell et al. (2019), ISv9 and LNLGv9 biases are closer to each other (in the v7 MIP, the LNv7 were biased high compared to ISv7). Additionally, the OCO-2 biases have decreased (to values between -1.0 and 1.0 ppm) with v9 compared to v7, where biases ranged between -1.5 and 1.5 ppm (Crowell et al., 2019). As observed here as well, IS and v9 have large positive biases over most of the European sites, which could indicate either an issue related to

the coarse resolution used by the transport models or (though this has not be studied yet) to the accuracy of TCCON retrievals over these regions (Crowell et al., 2019). The Caltech, Saga, and Tenerife sites show large underestimation in both the IS and v9 results across all models. As mentioned and observed in v7 MIP, differences between the Caltech and Edwards sites (which are very close each other) could be due to the location of Edwards over the mountains and Caltech is affected in the Los Angeles basin (Kort et al., 2012; Schwandner et al., 2017): the coarse resolution of models cannot differentiate the variability

of these two sites (Crowell et al., 2019; Schuh et al., 2021). This could also explain the underestimation observed over Saga





**Figure 13.** Normalized bias (top plots), normalized standard deviation (middle plots) and root-mean-square error (RMSE in ppm, bottom plots) by latitudes, for each models and for IS experiment (left panels) and LNLGv9 experiment (right panels).



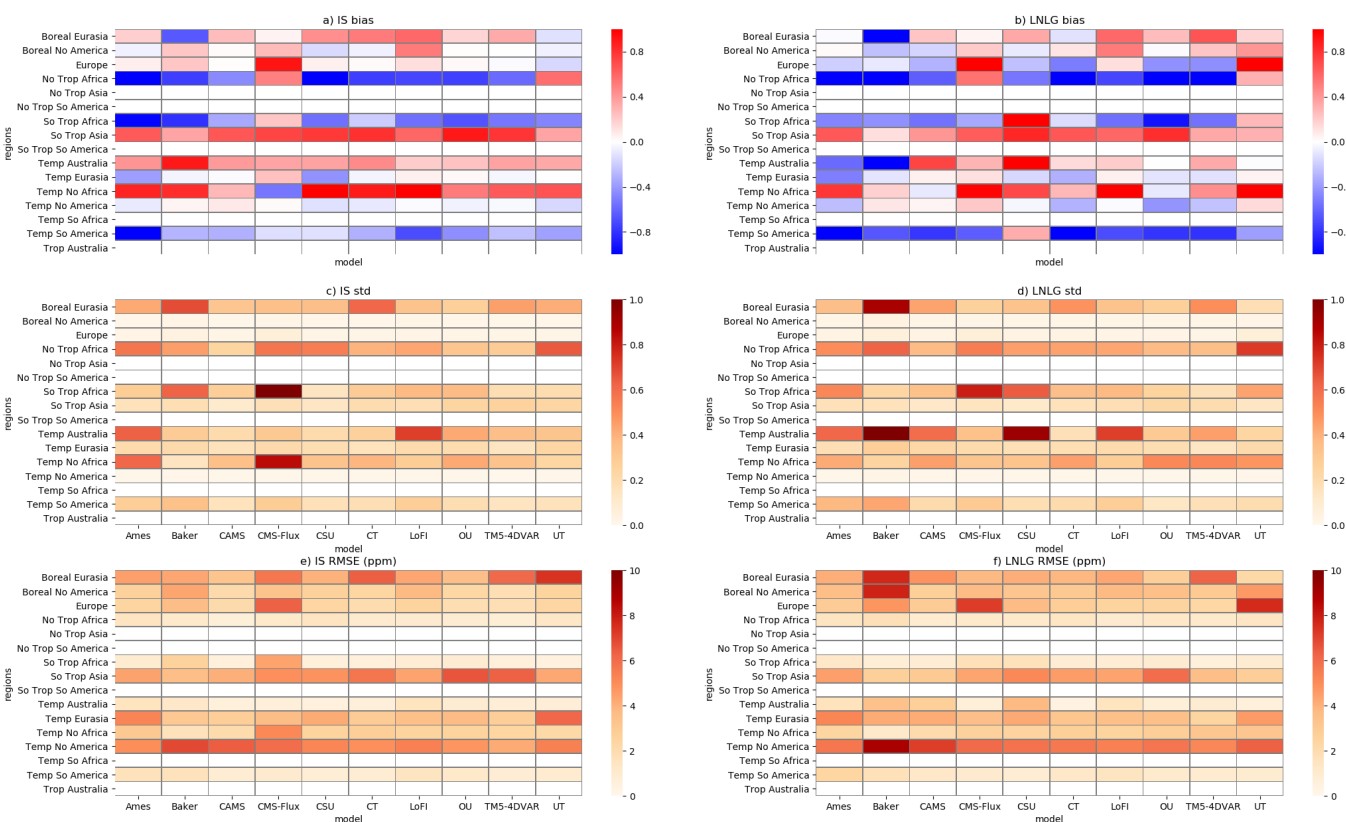

**Figure 14.** Same as Fig. 13 but for the MIP regions.

and Izaña (Tenerife Island), which are urban regions. Finally, most of sites over southern latitudinal bands are underestimated with LNLGv9 but slightly overestimated with IS, as also observed with the withheld *in situ* and ATom observations. However, Ascension Island, situated in the tropics, shows an overestimation of around 0.6 ppm for all models and for both IS and LNLGv9. The biases in v9 have decreased for Ascension Island compared to v7, where the biases were around 1.0 ppm for LN

and LG.

## 4  Discussions

We have analyzed inversions assimilating OCO-2 version 9 XCO$_2$ retrievals and compared them to *in situ* (IS) inversions, and to the OCO-2 v7 inversions. Our study is an update to the v7 MIP analysis of Crowell et al. (2019), which used OCO-2 v7 retrievals. In addition to comparing the LNLGv9 viewing mode inversion results to the IS ones, we also wanted to see if

differences existed between an ensemble of atmospheric inverse models using either LNLGv9 or LNv7 and LGv7. We remind





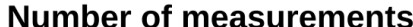

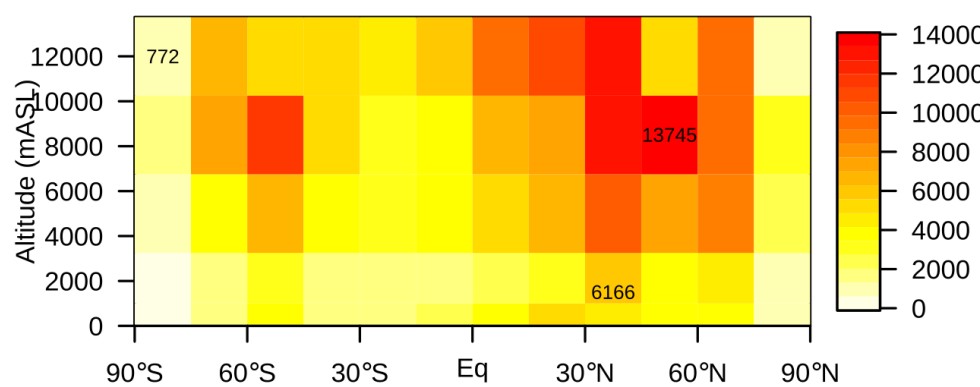

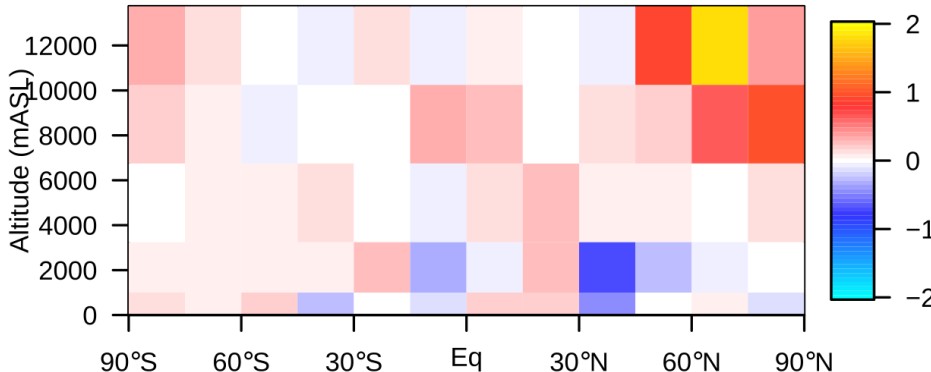

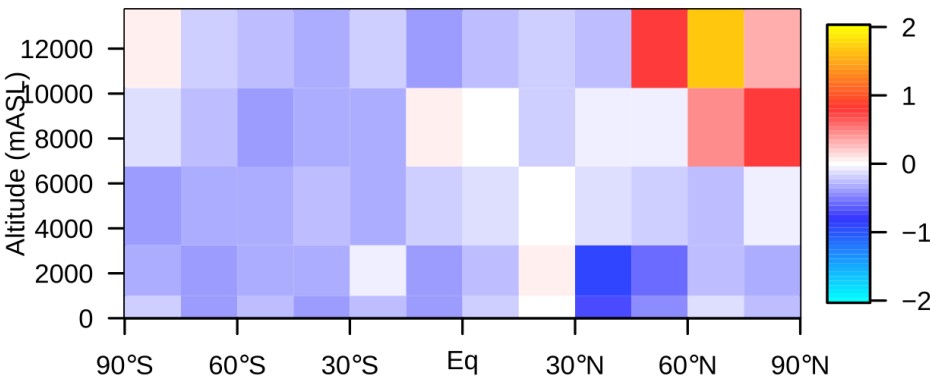

**Figure 15.** Comparison between ATom measurements samples and the MIP ensemble for the IS and LNLG experiments. The ATom samples have been binned into five altitude levels (going from the ground to 14 km). Top panel shows the number of measurements for each bin. Middle panel and bottom panel show, respectively, IS and LNLG experiment biases compared to ATom measurements. Biases are expressed in ppm $CO_2$.





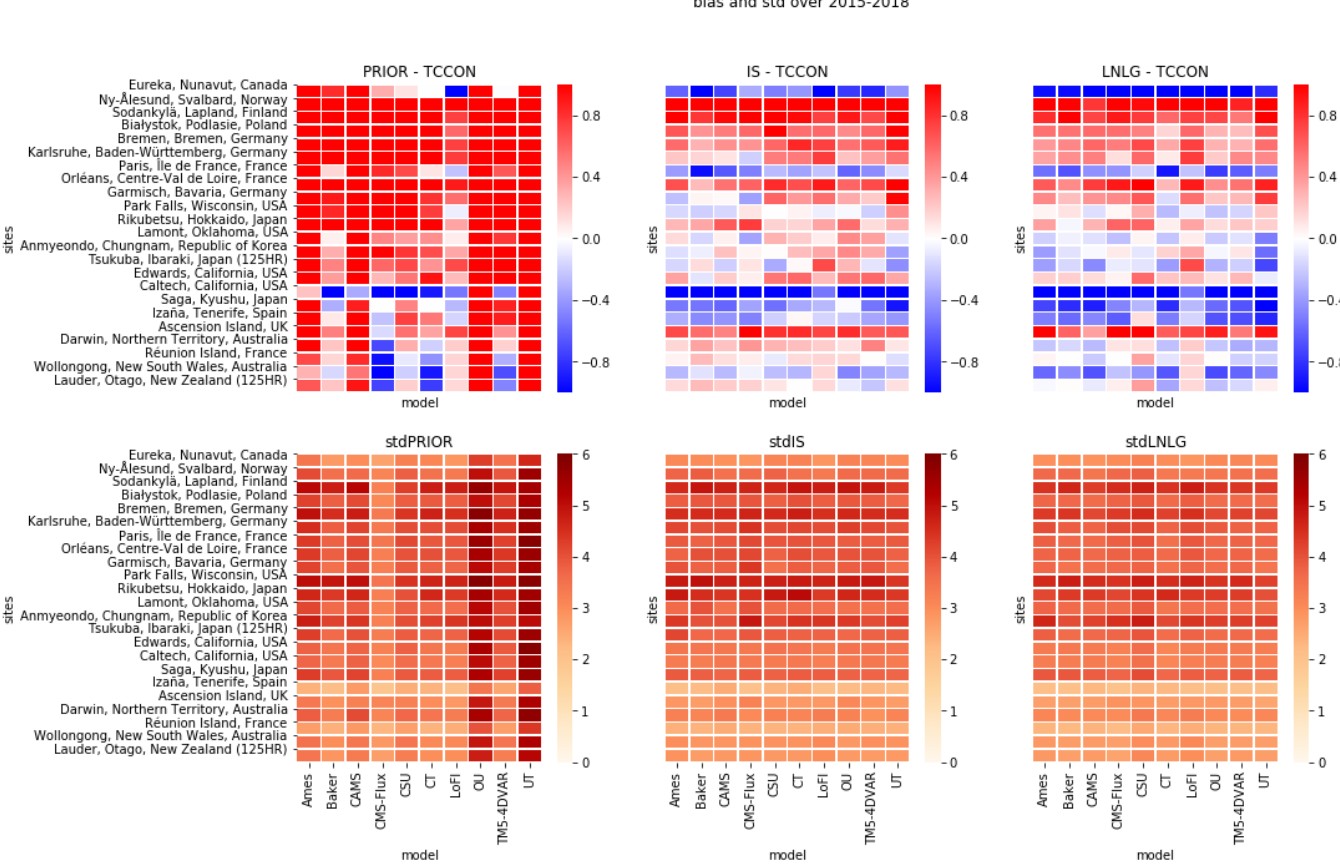

**Figure 16.** Comparison between TCCON data and the MIP ensemble for the IS and LNLG experiments. Top panels show experiments biases and bottom panels show standard deviation compared to TCCON sites. Biases and standard deviation are expressed in ppm $CO_2$. Prior are in the left sides, IS in the middle sides and LNLG in the right sides of the figure.

the reader that differences between $XCO_2$ in v7 and v9 are not related to meteorology which is fixed, but to spectroscopy and geolocation improvements, and better representation of aerosols (O'Dell et al., 2018). We have compared these different experiments, starting at the global scale, moving on to latitudinal scales, and finishing with regional scales. As expected, we did not find large differences between v7 and v9 at the global scale or latitudinal scale, except that the ensemble spread among the models was smaller with v9 than with v7. Transport model uncertainty is not expected to have changed dramatically since v7. This suggests that the reduction in the ensemble spread is likely related to a decrease in OCO-2 retrievals errors in v9 compared to v7.

However, posterior flux estimates between LNLGv9 and LNv7 and LGv7 differ for some regions. The key regions of interest that emerged from previous OCO-2 v7 MIP exercise and papers were in the tropics, including Tropical Africa, Tropical South America and Tropical Asia. The comparison between v7 and v9 for the Northern Tropical regions did not show large





differences, except a small increase of carbon sources with v9 compared to v7 in Northern Tropical South America, and a larger ensemble spread in Southern Tropical South America. Differences were also very small between v7 and v9 over Tropical Asia. However, we have observed a change in the carbon cycle between v7 and v9 for the Southern Tropical Africa, with carbon sink with v7 and carbon source with v9 during the El Niño period. Palmer et al. (2019) assimilated OCO-2 v7 land data and

GOSAT v7 data separately during the El Niño period and analyzed the posterior emissions over the pan-Tropical regions. With their inversions, they found carbon sources of 1.54 PgC/yr in 2015 and 1.72 PgC/yr in 2016 over the Northern hemisphere of the tropics. The carbon sources they found with OCO-2 v7 are similar to what we obtain with our OCO-2 v9 inversions. Palmer et al. (2019) found with v7 that the largest seasonal cycle of carbon fluxes in the tropics was over Northern Tropical Africa. This analysis would have probably been similar with v9. However, while they had -0.26 PgC/yr in 2015 and -0.18

PgC/yr in 2016 over the Southern Tropical latitudes, we observe around 0.75 PgC/yr in 2015 and 0.4 PgC/yr in 2016 with v9 (Fig. 8). Analyzing our inversions at regional scales, we saw that this opposite sign in emissions was coming from Tropical South Africa. In our Fig. 10, we have been able to observe a source of around 0.25 PgC/yr in 2015-2016, while v7 emissions were around -0.25PgC/yr for the same period, which is also what Palmer et al. (2019) found. They concluded that the largest carbon uptake was over the Northern Congo basin, situated in the Southern Tropical Africa MIP region. This result and the

difference between v7 and v9 suggest that the conclusion over the Southern Tropical Africa MIP region with v7 would have been different with v9. Liu et al. (2017) have assimilated OCO-2 v7 data to study the carbon cycle responses of the tropical regions to the 2015 El Niño and compared it to the 2011 La Niña. They found an respiration anomaly over Tropical Africa, with an increased release of carbon of 0.6 PgC/yr in 2015 compared to 2011. Gloor et al. (2018) did a similar study as Liu et al. (2017), but instead of comparing 2015 El Niño to 2011 La Niña, they compared 2015 El Niño to the 1998 El Niño.

Additionally, they did not assimilate satellite retrievals but rather *in situ* data from the NOAA surface station network. While their conclusions for Southern Tropical America and Southern Tropical Asia were similar to what Palmer et al. (2019) and Liu et al. (2017) found, they were surprised by their results over Tropical Africa. Contrary to their expectations, they found hot conditions in the Congo basin in February 2016 suggesting a release of carbon. Their results for the Congo basin (with source of carbon) were in opposition with the previous papers, which found sinks of carbon during the El Niño period of 2015-2016.

This anomaly over Africa was not expected, as Africa is generally not a tropical region affected by El Niño events. When they compared to total column carbon monoxide (CO) anomalies using MOPITT data, they found anomalous flux in Southern Tropical Africa, with a large CO release in February 2016. Finally, they also found a water deficit at the beginning of 2016 (weaker than over the Amazon). We can then see that this study assimilating *in situ* measurements is in agreement to what we observe with v9 and our IS inversions. The corrections in the retrievals (in going from v7 to v9) seem hence to be important

for $CO_2$ emissions estimates, particularly over tropical regions.

## 5  Conclusions

In this study, we compare an ensemble of inversion models separately assimilating *in situ* data, OCO-2 v7 LN and LG retrievals, and OCO-2 v9 LNLG retrievals, across the 2015-2018 period. Using the four years available with v9, in comparison to the two





with v7, we have been able to observe better the impact of El Niño period during 2015-2016 and the recovery period during

the 2017-2018 period, especially over the tropics. Additionally, the ensemble spread among the models assimilating OCO-2 v9 retrievals is smaller for almost all regions compared to the ensemble spread with OCO-2 v7 retrievals, meaning either the impact of the long period used with the different models and priors or a better agreement in emissions among the models and the impact of the v9 retrieval bias correction. We find at the global scale a good agreement overall between fluxes inferred from the v7, v9 and *in situ* data. However, differences are found at smaller scales over northern latitudes and particularly over

the tropics. While seasonality in the tropics differs significantly between the OCO-2 v7 and v9 compared to *in situ* results, the annual emissions show better agreement between the *in situ*, LNv7 and v9, except over Northern Tropical Africa and Northern Tropical Asia. As was observed with v7, v9 also shows a stronger source of carbon over Northern Tropical Africa than that observed with *in situ* data. This weaker source given by the *in situ* data seems to be balanced over Northern Tropical Asia with a larger out-gassing during the El Niño period that is not observed with the OCO-2 retrievals, as they show only sinks of carbon

there for the whole period. It is difficult to conclude where these sources come from, as there are few *in situ* observations over this region. Finally, we see, as previously mentioned in several studies, a carbon uptake (of around -0.25 PgC/yr) over Southern Tropical Africa using the v7 data, but a carbon release there using the v9 data of around 0.25 PgC/yr during the El Niño period. The *in situ* data also suggest a carbon release. This difference between v7 and v9 over Southern Tropical Africa seems to show the impact of retrieval bias corrections on the regional $CO_2$ fluxes, particularly in the tropics. This contradiction in the carbon

budget conclusion between the two sets of OCO-2 inversions requires further investigation over this African region.

Evaluation with the withheld data, ATom aircraft measurements, and TCCON retrievals suggests similarities in biases between the *in situ* data and LNLG v9 data, with a slight negative bias in the v9 OCO-2 data for almost all latitudes, particularly in the Southern Hemisphere and the tropics, where few evaluation data are available. Evaluation against TCCON shows also a reduction in retrieval errors with v9 ensemble models compared to v7.

Now that OCO-2 v10 retrievals are available, analysis and comparison between this new release and the two preceding ones presented in this study, should bring further flux information and comparison for the tropical regions regarding retrieval corrections.

**Data availability**

The surface gridded flux are available from https://www.esrl.noaa.gov/gmd/ccgg/OCO2_v9mip/. The *in situ* measurements are

available from https://www.esrl.noaa.gov/gmd/ccgg/obspack/.

**Appendix A:  Model information**

**A1    Ames**

The Ames inversion system used the transport model GEOS-Chem (Goddard Earth Observing System - Chemistry, Bey et al. (2001)), driven by meteorological parameters from the MERRA-2 reanalysis (Bosilovich et al., 2017) and run at a 4°x5° reso-





lution (further description provided in Philip et al. (2019)). Surface fluxes were optimized monthly. The prior land biospheric
$CO_2$ fluxes (also called prior NEE) used in the model setup comes from CarbonTracker CT2019 (Jacobson et al., 2020b)
based on CASA-GFED4.1s (Potter et al., 1993; Giglio et al., 2013). The prior ocean fluxes are from the unoptimized CT2019
Ocean Inversion Fluxes prior (OIF) product (Jacobson et al., 2007, 2020a). Finally, prior emission for fires are from GFED4.1s
(Giglio et al., 2013; Van Der Werf et al., 2017). Prior NEE flux error was calculated as the range of five different biospheric
$CO_2$ flux models (CT2019-CASA-GFED4.1s, CASA-GFED3, NASA-CASA, LPJ and SiB-4), scaled with 1.35 to represent
unaccounted uncertainty components, and keeping an upper bound of 5 times the absolute value of monthly prior NEE for each
surface grid box of the model. The scaling was also done to get global total NEE uncertainty values in agreement with Globlal
Carbon Project (GCP) estimates for the period of study (years 2015-2018). Oceanic prior flux error was assigned to be 5.0
times the absolute value of monthly oceanic prior fluxes in each surface grid box of the GEOS-Chem model. This scale factor
was also done to get global total ocean uncertainty values in agreement with GCP estimates for 2015-2018. For both NEE and
oceanic prior fluxes, no spatial or temporal correlations were considered.

## A2 Baker

The transport model used in the Baker simulations is the Parameterized Chemical Transport Model (PCTM, Kawa et al.
(2004)). The meteorology fields come from the MERRA-2 reanalysis: these have been coarsened to a resolution of 2°x2.5°
for the forward runs of the prior fluxes and to a resolution of 6.67°x6.67° for the assimilation of the measurements; the
vertical resolution has been coarsened to 40 levels in both cases by grouping together the original 72 layers in the upper
atmosphere. Biospheric priors come from the CASA-GFED3 (Potter et al., 1993; Randerson et al., 1997; van der Werf et al.,
2004) and include gross primary productivity (GPP), net ecosystem respiration, wildfires, and biofuel emissions. A multiple
of the respiration forward run is added onto the others to bring the global trend of $CO_2$ at Mauna Loa into agreement with the
observations across the 2014-2018 time span. The results shown are the mean of four separate inversions, each done using a
different air-sea flux priors: a) the NASA ocean biosphere model (NOBM, Gregg et al. (2003); Gregg and Casey (2007)), b)
the Takahashi et al. (2009) fluxes, c) the Landschützer et al. (2015) fluxes, and d) those same Landschützer et al. (2015) fluxes
with a sink of 0.95PgC/yr added across the Southern Ocean. Uncertainties in the priors are based on Baker et al. (2006), with
no correlations assumed between different weeks/grid boxes. Weekly fluxes are estimated using a variational data assimilation
scheme Baker et al. (2006).

## A3 CAMS

CAMS simulation used the global general circulation model LMDz with a spatial resolution of 1.9°x3.75° resolution and 39
vertical layers. The inferred fluxes are estimated in each horizontal grid point of the transport model with a temporal resolution
of 8 days, separately for day-time and night-time. ERA-Interim reanalysis meteorology fields are used. This inversion system is
part of the PyVAR-$CO_2$ configuration. Biospheric priors come from the climatology of the ORCHIDEE model version 4.6.9.5
(Krinner et al., 2005). Ocean prior are from the CMEMS (Denvil-Sommer et al., 2019). And fire prior are from GFED4 (Giglio
et al., 2013; Van Der Werf et al., 2017).





The biospheric prior errors are assumed, over land, to dominate the error budget and covariances are based on an analysis of mismatches with *in situ* flux measurements (Chevallier et al., 2006, 2012). Spatial correlations on daily mean NEE decay exponentially with a length of 500km; standard deviations are set to 0.8 times the climatological daily-varying heterotrophic respiration flux simulated by ORCHIDEE, with a ceiling of 4 $gC.m^{-2}$ per day. Over a full year, the total 1-sigma uncertainty for the prior land fluxes amounts to about 3.0 $GtC.yr^{-1}$.

Ocean prior uncertainty is defined as follows : (i) temporal correlations decay exponentially with a length of one month, (ii) unlike land, daytime and night-time flux errors are fully correlated, (iii) spatial correlations follow an e-folding length of 1000 km; standard deviations are set to 0.1 $gC.m^{-2}$ per day.The global air-sea flux uncertainty is about 0.5 $GtC.yr^{-1}$.Land and ocean flux errors are not correlated.

## A4  CMS-Flux

Carbon Monitoring System (CMS)-Flux used a four-dimensional variational (4D-Var) inversion approach with the model GEOS-Chem. The model is driven by the Goddard Earth Observing System version 5 of the NASA Global Modeling Assimilation Office (GEOS-FP) meteorology and runs at a 4°x5° resolution. Net Biospheric Exchange (NBE) prior has been constructed using the CARDAMOM framework (Carbon Data Model Framework, Bloom et al. (2016)). The CARDAMOM data assimilation system explicitly represents the time-resolved uncertainties in the NBE. The prior estimates are already constrained with multiple data streams accounting for measurement uncertainties following a Bayesian approach similar to that used in the 4D-variational approach. CMS-Flux simulation use the CARDAMOM setup as described by Bloom et al. (2016); Anthony Bloom et al. (2020) resolved at monthly timescales; data constraints include GOME-2 solar-induced fluorescence (Joiner et al., 2013), MODIS Leaf Area Index (LAI), and biomass and soil carbon. In addition, mean GPP and fire carbon emissions from 2010 - 2017 are constrained by FLUXCOM RS+METEO version 1 GPP (Tramontana et al., 2016; Jung et al., 2017) and GFEDv4.1s (Randerson et al., 2012), respectively, both assimilated with an uncertainty of 20%. The Olsen and Randerson (2004) approach has been used to downscale monthly GPP and respiration fluxes to 3-hourly timescales, based on ERA-interim re-analysis of global radiation and surface temperature. Fire fluxes are downscaled using the GFEDv4.1 daily and diurnal scale factors on monthly emissions. Posterior CARDAMOM NBE estimates are then summarized as NBE mean and standard deviation values. The NBE from CARDAMOM shows net carbon uptake of 2.3 GtC/year over the tropics and close to neutral in the extratropics. The year-to-year variability (i.e., interannual variability, IAV) estimated from CARDAMOM from 2010 –2017 is generally less than 0.1 gC/m 2 /day outside of the tropics. Because of the weak interannual variability estimated by CARDAMOM, the same 2017 NBE prior is used for 2018. CARDAMOM generates uncertainty along with the mean state. The relative uncertainty over the tropics is generally larger than 100%, and the magnitude is between 50% and 100% over the extra-tropics. We assume no correlation in the prior flux errors in either space or time.

## A5  CSU

CSU simulation used GEOS-Chem model with a Bayesian technique at a 1°x1° resolution. The model is driven by MERRA-2 meteorology fields. Biospheric prior emissions are based on the Simple Biosphere Model Version 4 (SiB4, Baker et al. (2013))





at a 1x1 resolution grid. SiB4 is a land surface model that predicts vegetation and soil moisture , land surface energy and terrestrial carbon cycle. It used the carbon fluxes to determine biomass above and below the ground. Prior ocean are based on CT2019 OIF and fire priors on GFED4.

## A6   CT

The CarbonTracker (CT) simulations presented here closely follow the methods used for CT2019B (Jacobson et al., 2020b), except that only one set of biosphere, wildfire, and oceanic emissions data were used. For the present experiments, first guess land carbon emissions were provided by GFED4.1s (Giglio et al., 2013; Van Der Werf et al., 2017) and first-guess ocean emissions come from Jacobson et al. (2007). CT inversions optimize surface fluxes by estimating weekly scaling factors multiplying first-guess emissions for 156 ecoregions covering the globe. The optimization scheme is an ensemble Kalman

filter with a 12-week assimilation window. Atmospheric transport is simulated by TM5 (Krol et al., 2005) with meteorology from the ERA-Interim reanalysis (Dee et al., 2011) with a global resolution of 3° longitude by 2° latitude and a 1°x1° zoom region over North America.

## A7   LoFI

While the OCO-2 MIP v7 (Crowell et al., 2019) was composed of models assimilating OCO-2 retrieval v7 or *in situ* data,

the OCO-2 MIP v9 has in addition the model LoFI (Weir et al. (2020)) which is an integrated Earth system model with data assimilation capabilities. The Low-order Flu Inversion (LoFI,(Weir et al., 2020)) reproduce atmospheric and oceanic growth rates from a group of remote sensing carbon fluxes. The model used is Heracles 4.0 GEOS GCM, including MERRA-2 meteorological inputs and a 0.5°x0.625° resolution grid with 72 vertical levels. Biospheric fluxes have a 3-hourly timestep while all other fluxes have a daily timestep. Biospheric and biofuel priors come from the CASA-GFED3. Biomass burning

priors are based on the Quick Fire Emissions Dataset (QFED, Darmenov and Silva (2015)). QFED is based on MODIS fire radiative power estimates using the technique from GFAS (Kaiser et al., 2012). Finally, oceanic priors are an extension of Takahashi et al. (2009) climatology product used with NOAA zonal-mean surface $CO_2$ and MERRA-2 wind speed. This approach is also used in NOAA CarbonTracker system.

## A8   OU

OU simulation used the chemistry transport model TM5 (Krol et al., 2005) with a 4DVAR assimilation algorithm. The model has been run at a 4°x6° resolution with 25 vertical layers. ERA-Interim meteorology fields are used here as well. Initial conditions are provided from CarbonTracker. Prior oceanic were constructed from Takahashi et al. (2009). Biospheric priors are based on CT2019 CASA-GFED3 and fire prior from GFED3. Uncertainties are derived from different climatological fluxes. Exponential spatiotemporal correlation is assumed for the uncertainty in the prior flux. For the oceanic component, length is

of 1000km and timescales of 3 weeks, while for the terrestrial component, length and timescale are of 250 km and 1 week.



## A9 TM5-4DVAR

TM5-4DVAR simulation used the same transport model as OU simulation with same meterological fields but at a 2°x3° resolution grid. A climatological average of CT2019 oceanic fluxes estimates constrained prior oceanic fluxes. Biospheric prior are taken form SiB CASA GFED4 (Van Der Velde et al., 2013) and fire prior from GFED4 (Randerson et al., 2012). The uncertainties on the biospheric prior are fixed to 0.5 times the heterotrophic respiration from SiB CASA. For the oceanic prior, the uncertainties are fixed at 1.57 times the absolute flux at each grid cell and time step. Same correlation, length and timescale of OU simulation are assumed and used in this TM5-4DVAR simulation.

## A10 UT

The Geos-Chem model has been used in UT simulation, driven by assimilated meteorological observation from GEOS-FP and used with the 4D-Var assimilation algorithm. The model is runs at 4°x5° resolution with 47 vertical layers. More information can be found in Deng et al. (2016). Fire priors are based on GFED4. Oceanic prior fluxes are based on the monthly climatology of Takahashi et al. (2009). And finally, biospheric prior are based on a 3-hourly fluxes from the Boreal Ecosystem Productivity Simulator (Chen et al., 2012). Annual terrestrial ecosystem exchange are assumed neutral in each grid box (Deng and Chen, 2011; Deng et al., 2014). Optimized scaling factors are estimated with a monthly temporal resolution. Uncertainty is assumed of 38% for the fire emissions in each month and each model grid box, while it is of 44% for the ocean emissions and of 22% for terrestrial emissions.

*Author contributions.* H.Peiro completed the analysis of the results, wrote the manuscript and produced the figures. S.Crowell, D.Baker, A.Schuh, S.Basu, A.Jacobson, F.Chevallier, F.Deng, J.Liu, M.Johnson, S.Philip, and B.Weir provided surface flux estimates. D.Baker provided the OCO-2 10 s averages. A.Jacobson provided *in situ* observations for assimilation, the withheld data for evaluation and provided ATom figures. S.Basu produced the quality-controlled TCCON time-averaged data. C.O'Dell, D.Crisp and A.Eldering provided feedback on the manuscript. S.Crowell, D.Baker, A.Schuh, A.Jacobson, F.Chevallier, F.Deng, J.Liu, M.Johnson, S.Philip, I.Baker, and B.Weir provided review and comment on the final paper.

*Competing interests.* The authors declare that they have no conflict of interest.

*Acknowledgements.* The TCCON data were obtained from the TCCON Data Archive hosted by CaltechDATA at https://tccondata.org. We thanks TCCON PIs for the TCCON measurements at Eureka, Ny-Ålesund, Sodankylä, Białystok, Bremen, Karlsruhe, Paris, Orléans, Garmisch, Park Falls, Rikubetsu, Lamont, Anmeyondo, Tsukuba, Edwards, Caltech, Saga, Izaña, Ascension Island, Darwin, Réunion Island, Wollongong, Lauder. Eureka measurements are made by the Canadian Network for the Detection of Atmospheric Change (CANDAC) and in part by the Canadian Arctic ACE Validation Campaigns. They are supported by the Atlantic Innovation Fund/Nova Scotia Research In-



novation Trust, Canada Foundation for Innovation, Canadian Foundation for Climate and Atmospheric Sciences, Canadian Space Agency,
Environment Canada, Government of Canada International Polar Year funding, Natural Sciences and Engineering Research Council, North-
ern Scientific Training Program, Ontario Innovation Trust, Ontario Research Fund and Polar Continental Shelf Program. Observations for
Białystok are funded byt the European Union (EU) projects InGOS and ICOS-INWIRE, and bu the Senate of Bremen. Local support for
Bremen and Ny-Ålesund are provided by the EU projects InGOS and ICOS-INWIRE (26188, 36677, 284274, 313169 and 640276), and
by the Senate of Bremen. Orléans observations are supported by the EU projects InGOS and ICOS-INWIRE, by the Senate of Bremen and
by the RAMCES team at LSCE. The Paris and Réunion Island TCCON sites have received funding from Université Pierre et Marie Curie,
the French research center CNRS, the French space agency CNES, and Rtégion Île-de-France. Garmisch funding was provided by the EC
within the INGOS project. Park Falls, Lamont, Edwards and Caltech TCCON site have received funding from National Aeronautics and
Space Administration (NASA) grants NNX14AI60G, NNX11AG01G, NAG5-12247, NNG05-GD07G, and NASA Orbiting Carbon Obser-
vatory Program. They are supported in part by the OCO and OCO-2 series project. The TCCON station at Rikubetsu, Anmeyondo and
Tsukuba are supported in part by the GOSAT series project. Darwin and Wollongong TCCON stations are funded by NASA grants NAG5-
12247 and NNG05-GD07G and supported by the Australian Research Council (ARC) grants DP140101552, DP110103118, DP0879468 and
LP0562346. Lauder TCCON site has received funding from National Institute of Water and Atmospheric (NIWA) Research through New
Zealand's Ministry of Business, Innovation and Employment. We also acknowledge the NASA ATom aircraft-campaign data from the NASA
Ames Earth Science Project Office (https://espo.nasa.gov/atom).



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
