# Peer review of "Four years of global carbon cycle observed from OCO-2 version 9 and *in situ* data, and comparison to OCO-2 version 7"

_Atmospheric Chemistry and Physics, 2021_

## Referee Comment (RC1)

**Interactive discussion on "Four years of global carbon cycle observed from OCO-2 version 9 and in situ data, and comparison to OCO-2 v7" by Peiro et al., (2021).**

The manuscript written by Peiro et al., (2021) is an update of the study made by Crowell et al., (2019), et al., (2019), who assessed the annual and monthly ensemble mean flux derived by nine different global inversions using the assimilation of OCO-2 satellite data (version 7) and in situ data for the period 2015-2016. Helene et al., 2019 used the findings found in Crowell et al. (2019) and compared them with the ensemble median flux of 10 different global inversions based on the assimilation of OCO-2 (version 9) land nadir and land glint (LNLG) data. Further, the authors also analysed fluxes for a more extended period 2015-2018, using both in situ and OCO-2 satellite data separately. In general, the manuscript is well-written, and the discussion and conclusions are interesting. However, some parts in the methodology and results sections need revision and clarifications. I would recommend this paper for publication once the authors have addressed all the questions described below.

**General comments:**

Line 15. "However, the lack of data in the tropics limits our conclusions and the estimation of carbon emissions over tropical Africa require further analysis". It would be good If the author can provide spatial maps of LNLG OCO-2 across the world for the studying years to see how bad the coverage is in Tropics. (Maybe show these maps in the appendix of the manuscript).

Line 137. "This bias correction in v9 allows a more uniformly distribution of XCO2 over regions of interest, decreasing the standard deviation to 0.74 ppm compared to v8, which was of 1.35 ppm." Please indicate that this reduction was found in TCCON Lauder, New Zealand (Fig.13).

Line 147-151 Interestingly, the authors accounted for correlations between the model-data mismatch error for the 10 second average of OCO2 soundings. However, the description of how it was calculated is not clear. It would be great if the author could provide more details in this section of the manuscript.

Line 159. "Both resulting uncertainty is then passed through the formula to account for error correlation." Which formula? It is hard to understand what the author did to calculate the observation uncertainties. Please provide more details in this section.

Line 160. "An additional term is added in quadrature to account for transport model errors. This model error term is computed from the difference between the CO2 concentrations computed by the TM5 and GEOS-Chem models when both are driven by the same realistic surface CO2 fluxes, after the annual mean difference filed is subtracted off".

The author referenced Basu et al., (2018) when they described the transport model error. In Basu et al., (2018), the model part error was calculated by considering the difference between a suite of inverse models optimised with in situ data and OCO-2 retrievals? Did the author do something similar here? Please clarify.

Line 240. For LNLGv9 inversions, the available 10s OCO-2 retrievals were averaged and compared to TCCON observations". I think the author means prior and posterior XCO2 simulated retrievals were averaged and compared to TCCON?

Line 257-258. Please indicates that Figure 5 represent only the median annual flux and the monthly median flux. In Cronwell et al., 2019, the authors discussed the annual and monthly mean flux, not the median. Could you indicate why you decided to compare the median and not the mean of the flux?

Line 261-262: "However, the peak sinks during the Northern Hemisphere growing season (from May through September) are slightly larger with OCO-2 v7 than with OCO-2 v9". Something is missing here. In this section, you discuss global flux estimates, and at the same time, the author discusses fluxes in the Northern hemisphere. I cannot see the Northern Hemisphere Figure for this section.

The author includes Lofi inversion in the paper. However, there are no comments about their results. For example, Lofi seems to agree with the prior median estimate over the Northern Extra Tropics but not with IS and LNLGv9 assimilated flux.

Line 310- 313. "In the Southern Extra-Tropics, the authors mention that the sink flux estimate with OCO-2v9 is stronger than estimates made with IS, and v7, mainly because the ensemble spread with v9 is larger than with v7." I am a bit surprised that fluxes assimilated with v9 have a larger spread than v7. OCO-2 v9 data supposed to have lower biases than v7. It seems that the error in the transport model might be the consequence of the large spread in the inversion with v9.

Line 314-318. Over the Tropics, the authors mention that posterior flux estimate (LNLG) is quite different to the prior estimate (Fig 6.c), but not comments are provided. How reliable can be the posterior estimate over the Tropics knowing that OCO-2 retrievals can be biased due to cloud coverage during the wet season and aerosol from biomass burning during the dry season, as the author mentioned?

Could you explain a bit more about the dipole in northern Africa? You mentioned several studies, but a better explanation is needed.

Fig 8 (NH Tropics) and (SH Tropics) LoFi monthly seasonally seem to be offset? Could the author explain why LoFi might not be capturing the seasonally over these latitudes bands?

Line 412. Alternatively, the OCO-2 inversions could be dominated by savanna seasonality (Baker et al., 2021a in prep). I cannot find (Baker et al., 2021a) in the reference list, and the reference cited (Baker et al., 2021) does not mention anything about the savanna seasonality but how to calculate the error correlations in OCO-2 data.  Please provide the correct reference.

Line 426-. "Over the southern hemisphere, a large underestimation of the ensemble mean of LNLGv9 appears compared to the observations". I am surprised with these findings; I would have thought that the spatial coverage that OCO-2 data in the southern hemisphere would improve the results compared to IS. I am also surprised by the significant difference between LNLGv9 and IS biases in SH. Do you know how large are LNLGv9 OCO-2 biases in SH compared to v7? You mentioned in the introduction that biases in LNLG v9 were reduced considerably compared to v8 and v7. Is it possible to provide a Table in the appendix with no normalized bias?

Line 466. It is possible then that this excess of concentration, in both experiments, reflects the initial conditions of the inversion. Is there any way to test this?

Fig 13 and Fig 14. I don't understand why the author normalized the biases. It is clear that RMSE is large at some latitudinal bands and TCCON sites, suggesting that raw biases might also be high. I would also consider adding this information to the manuscript.

Line 471-474. "Compared to the evaluation of ISv7 in the study of Crowell et al. (2019), ISv9 and LNLGv9 biases are closer to each other (in the v7 MIP, the LNv7 were biased high compared to ISv7). Additionally, the OCO-2 biases have decreased (to values between -1.0 and 1.0 ppm) with v9 compared to v7, where biases ranged between -1.5 and 1.5 ppm (Crowell et al., 2019)". I don't understand why the authors say that OCO-2 biases have decreased compared to Crowell et al., 2019. Did Crowell et al., 2019 normalize the bias? If not, I don't quite understand the comparison.

Line 474-476. "…to the accuracy of TCCON retrievals over these regions (Crowell et al., 2019). I think that biases might likely be associated with biases to satellite observations than TCCON bias. Besides, I dont think that Cronwell et al., 2019 is not a good reference for talking about the accuracy of TCCON. XCO2 TCCON retrievals can contain airmass-dependent biases, which are corrected using the method described by Wunch et at.. Just wondering how wrong could be this correction to cause the posterior concentration biases seen here.

Line 476. Could the authors provide the location of the TCCON sites in a map instead of Table 3? It would be better for the reader to see where Caltech, Saga and Tenerife TCCON sites are located. I had to look at their locations from other manuscript.

The Caltech, Saga, and Tenerife sites show large underestimations in the IS and v9 results across all models. Isn't it bad to find a large underestimation at Caltech for the reliability of the fluxes estimated in North America? Is the location of Caltech a coastal site in the model? If so, it might strongly affect by ocean fluxes where not OCO-2 observations were assimilated.

Line 480. Another possible explanation of the underestimation observed over Saga and Izaga is that these small islands are strongly influenced by ocean fluxes, where the assumed uncertainties are small compared to land.

The biases in v9 have decreased for Ascension Island compared to v7, where the biases were around 1.0 ppm for LN and LG. Are these results compared to TCCON biases presented in Cronwell 2019? If so, I don't understand why the authors compare standardized bias against not standardized biases?

Line 495. "Transport model uncertainty is not expected to have changed dramatically since v7. This suggests that the reduction in the ensemble spread is likely related to a decrease in OCO-2 retrievals errors in v9 compared to v7". Are the modellers that participate in OCO-2 MIP have trying to consider improvement in the transport modelling (at the surface)?

Line 525-526. "When they compared..." who is they? I think the reference is missed here.

In the discussion section, I think it is also important that the author indicates why the assimilating OCO2 LNLGv9 data over the Tropics shows a stronger seasonality than prior fluxes. For the reader, it would be good to know why prior fluxes derived from ORCHIDDE or CARDAMON are not capturing well the seasonality. What could be wrong with these models?

Line 552 -554 "…with a slight negative bias in the v9 OCO-2 data for almost all latitudes, particularly

in the Southern Hemisphere and the tropics, where few evaluation data are available". Here the author mentions "slight negative bias in the v9 OCO-2 data", however in lines 426-427, they write, "Over the southern hemisphere, a large underestimation of the ensemble mean of LNLGv9 appears compared to the observations". This seems a bit contradictory.

**Editorial and minor comments:**

Title: Four years of global carbon cycle observed from OCO-2 version 9 and in situ data, and comparison to OCO-2 v7. As a personal opinion, I would write version 7 instead of v7 to be consistent with "version 9".

Line 121. "Inversions assimilating OCO-2 ocean retrievals produced unrealistic results with annual global ocean sinks higher of $2.6 \pm 0.5$ GtC.yr−1 compared to the state-of-the-art estimated in Le Quéré et al. (2018)." Could you quote the ocean sink estimated in Le Quéré et al. (2018).

Line 136. (Kiel et al. (2019)). Remove parenthesis to 2019.

Line 151. "Details of the form and derivation of these average uncertainties may be found in the 'constant correlation' section of Baker et al. (2021).", Please be consistent with the quotation mark "constant correlation".

Line 137. "This bias correction in v9 allows a more uniformly. Replace the word uniformly by uniform.

Line 184. There is an extra point. Please remove.

Line 241-242 Figure3. Please add the y-axis description (right legend).

Line 311. "However, in contrast to NHExt". In this line, you need to define define NHEx.

Line 305: The Southern extra-tropics. Capital letter?.

Line 306. Correct the word "signifcaintly" to significantly.

Line 398. For the monthly emissions of the Southern Tropical regions (Fig. 10), we can see the strong impact of El Niño in fall 2015 over Southern Tropical Asia in the larger emissions (with a maximum of $0.35 \pm 0.01$ PgC/yr) given by all the inversions compared to the rest of the period. Please re-write, it is hard to understand.

Line 400. This large fall 2015 mainly come from Indonesian fires. I would write: The large emissions from Southern Tropical Asia (Fig.10.f) primarily come from Indonesia fires.

I would also remove the cyan and green lines from all plots (Fig.5 to Fig.10) that show monthly median fluxes for 2018-2019. If there is no data there, I don't think it is needed to show that the median is zero.

Line 605. … is about 0.5 GtC.yr−1.Land (need space).

---

## Author Response (AR2)

Reply to Reviewer #1

This is potentially an interesting paper that provides an update about the status of OCO-2 data. The authors use an inversion ensemble to compare against a range of independent data. Using an ensemble of models is a strength of this study but since the models do not use common prior information this reviewer was unable to understand how OCO-2 improved the performance of individual models. The manuscript would benefit greatly from a better exposition of the performance of individual models from the standpoint of error reductions. This would make it easier for other readers to fully appreciate the results of this study. Major and minor comments are listed below.

We are grateful for the comments of Reviewer #1 and for taking time to review our manuscript. We answered below the comments with information on page and line numbers that have been changed in the manuscript when necessary.

1- MAJOR. The MIP design claims to mimic past model intercomparison projects. Aside from using a common prior for the fossil fuel emissions, individual modeling groups were free to choose all other model features, e.g. biospheric priors, uncertainties for data, errors associated with model transport, etc (Table 1). These differences will impact posterior flux estimates. The authors argue that variations among inversion systems are considered beneficial for the purpose of characterizing flux uncertainty, but this reviewer would argue that some portion of this variation in the ensemble is unnecessary and a reflection of the design of the MIP. Obviously, it is too late to redo this ensemble experiment but given the expertise and complexity of the inversion systems it would be incredibly useful to report individual model prior and posterior uncertainties. Did some models use stricter constraints? Do some models follow their prior more than others? Did some models overfit data? How well did the individual model fit the net fluxes before the fossil fuel component was removed? Were the net $CO_2$ fluxes consistent with NOAA atmospheric $CO_2$ growth rate estimates? It would be useful to understand the basics of individual model performance before embarking on more elaborate data analysis. Otherwise, it is difficult to understand what knowledge has been gained from this experiment.

Previous model intercomparison experiments – for $CO_2$ and other long-lived species – have had limited control on individual model setups. The Global Carbon Atlas (http://www.globalcarbonatlas.org/en/content/atmospheric-inversions) infers the state of the

carbon cycle over the past decades from a diversity of $CO_2$ inverse models with no common elements, post-hoc corrected for the assumed fossil fuel emissions (Peylin et al., 2013). The Global Carbon Project (GCP) routinely uses model ensembles, both top-down and bottom-up, to construct $CO_2$ and $CH_4$ budgets (Friedlingstein et al., 2020; Saunois et al., 2020). The model ensembles used by the GCP for this purpose also have very limited or no commonality imposed on them. This is by design, since inter-model differences are likely to be a better estimator of the true uncertainty in our knowledge compared to the uncertainty estimate from any single model. It is true that in such model ensembles, not all models "perform" well, i.e., the ensemble is not well calibrated. Despite this limitation, as top-down models have continued to develop, they have started converging on certain key aspects of the global carbon cycle (Gaubert et al., 2019).

Compared to the top-down model ensembles mentioned above, the OCO2 MIP ensemble is more tightly controlled. They all use the same fossil fuel emissions, the same set of OCO2 retrievals with identical errors for assimilation, and the same set of in situ $CO_2$ observations with recommended errors used by most models. This level of control is, to the best of our knowledge, unprecedented among published top-down $CO_2$ model intercomparisons. The only aspects of the MIP not controlled are (i) the specification of the prior flux and its covariance, (ii) the assimilation scheme, and (iii) the atmospheric transport model. We argue that (ii) and (iii) provide a necessary diversity in model setup to gauge the true uncertainty in our results and emphasize that they are strengths rather than weaknesses of the MIP protocol. Indeed, inter-model differences due to (ii) and (iii) are often larger than due to the assimilated data (Chevallier et al., 2014), and it is therefore necessary to have some diversity in those aspects to represent the true posterior uncertainty. As for (i), specification of the prior flux and its uncertainty are tied very closely to the model setup and is therefore hard to standardize. The prior flux uncertainty, in particular, is incredibly difficult to standardize in reduced rank techniques such as 4DVAR and EnKF even among frameworks that use the same transport model (Babenhauserheide et al., 2015), and to the best of our knowledge no one has done that successfully. It is in theory possible to standardize the prior biospheric and oceanic fluxes, which is a direction future GCP $CO_2$ and $CH_4$ inversion efforts are taking. However, we would argue that the spread among bottom-up flux models (e.g., the TRENDY and OCMIP ensembles) is greater than the prior uncertainty one would construct based on any one model. Therefore, having a diversity of prior fluxes is also a strength of our protocol.

We consequently decided to add a sentence in our manuscript page 4 line 80: "MIP models are more strictly controlled or have more common elements than previous $CO_2$ inverse model intercomparisons such as Peylin et al (2013)".

The reviewer raises a very good point that given the diversity of model setup, not all models are equally good, and it is important to evaluate the performance of individual models before drawing conclusions about the carbon cycle. As part of the MIP, we did collect extensive validation statistics against non-assimilated in situ and column data. These metrics are available at https://gml.noaa.gov/ccgg/OCO2_v9mip/, both as global aggregate statistics and at a site level. However, we stopped short of using those statistics to weight models for two reasons. First, there is to date no consensus on how to weight the MIP models. We have tried several approaches from Bayesian model averaging (Massoud et al., 2020) to analysis of variance (Cressie et al., 2021), but have yet to find a satisfactory scheme. This remains an area of active research. Second, we suspect and hope that the MIP fluxes will be used by a variety of studies focusing on different regions on the globe. While it may be possible to derive global statistics for the models' performance, those statistics may not correlate with how well the models do over individual regions. We have therefore left it to the discretion of future users to choose MIP models wisely for their specific studies, aided by the metrics published at https://gml.noaa.gov/ccgg/OCO2_v9mip/.

Lastly, we would like to address a few specific questions raised by the referee which we think have already been answered in the manuscript:

- "Did some models use stricter constraints?" On the data, all models used the same constraints. On the prior fluxes, the models had different constraints.
- "Did some models overfit data?" Fit statistics are available at https://gml.noaa.gov/ccgg/OCO2_v9mip/co2tser.php. To the best of our knowledge, none of the models overfit the data.
- "How well did the individual model fit the net fluxes before the fossil fuel component was removed?" We do not understand this question, since none of the models try to *fit* the net fluxes.
- "Were the net $CO_2$ fluxes consistent with NOAA atmospheric $CO_2$ growth rate estimates?" Over the four years considered in the study, all the models fit the NOAA atmospheric $CO_2$ growth rate.

2- MAJOR. Differences in median prior fluxes between Crowell et al and this study are not explained. It is unfortunate that this group did not use consistent fluxes for their ensemble study or use the same fluxes used by Crowell et al. Figure 5 shows a wide range of values being used. Again, as described above, it is impossible to see which models track their priors more than others. This is particularly relevant for the analysis of the IS data in the tropics where coverage is sparse.

In our paper, we are not evaluating the different models, but the retrieval version of OCO-2 used in a model ensemble. As previously mentioned, information of each prior used and how they compare to the LNLG and IS posterior of each model are available in the OCO-2 MIP website. While we understand the desire for consistent priors across v7 and v9, it is worth noting that the MIP activity is largely supported by leveraging individual and separately funded research projects and thus is subject to necessarily changing models describing both the biosphere and atmospheric transport.

If we look at the figure on the website and our following figures for each individual model between their priors, IS and LNLG posterior over the tropics, we can see that neither IS nor LNLGv9 for each models follows their priors over the long period of 2015-2018. This can be observed for both the seasonality and the magnitude of the fluxes.

[Figure]

*Fig 1. Monthly mean fluxes in PgC/month (bottom side) and annual mean flux in PgC/yr (top side) of Tropics Land from 2015 through 2018 among the different models for Prior (left side), IS (middle) and LNLGv9 (right side).*

We mentioned this in our manuscript page 18 line 345: "We can also observe that the IS ensemble spread does not deviate from the prior spread. Neither IS nor LNLGv9, for each model, follows their priors in the tropics during the period of study (not shown here)."

3- MAJOR. Section 3.3 is largely superficial from the perspective of understanding reported flux estimates. Some versions of the fluxes agree with others versions... This is only interesting, if you tell the readers why you think this is important or relevant. This reviewer is surprised by the result over North America. Given this is a continental with a wealth of independent data this result is worrisome. Are there any model outliers that would explain some of the variations that are being reported? This needs more thought. Similarly, over Europe they are implying that their data have a large carbon sink but do not explicitly say it. Follow-up studies suggested this might not be correct (not cited) so Peiro et al could be a useful addition to the broader debate. The authors go on to suggest the enhanced summertime uptake might be due to a dipole between Europe and northern Africa and cite Houweling et al (2015). That's lazy. What did Peiro et al find? They have substantial computational machinery at their disposal. Any consensus among their models about this dipole? The authors have a great opportunity with their ensemble to do some insightful analysis.

For North America, there is a larger spread among the models with IS than with LNLGv9. Some models observed a weaker sink with IS than with LNLGv9 (information from the ensemble spread and median values in Fig. 7 of the manuscript). Schuh et al., 2019 and Schuh et al., 2021 demonstrated large differences in the flux inversions primarily over the Northern Midlatitudes mainly due to transport. This has not been investigated in our study, but could be the first reason for differences observed for Northern America (Fig. 7.a) and North Asia (Fig. 7.e).

It was mentioned in the manuscript (lines 339) that larger sink over Europe is observed with v9 than with v7. We can observe with the monthly fluxes that this sink is strong during the summer months. We were not interested in evaluating or comparing the different models but were interested in looking only to the ensemble mean.

But if we examine the individual models; we can also see the dipole between Europe and Northern Africa among each model (see Fig. 2) with sinks over Europe balanced with sources over Northern Africa.

[Figure]

*Fig 2. Monthly mean fluxes in PgC/month (bottom side) and annual mean flux in PgC/yr (top side) from 2015 through 2018 among the different models for LNLGv9. Fluxes over Europe are on the left side of the figure and Northern Africa is on the right side of the figure.*

4- MAJOR. Section 3.3 continues with a discussion about northern Asia and weaker/stronger sinks that could be due to different amounts of data being assimilated. Surely, the authors could find this out with their analysis. This reviewer would like to see more definitive statements based on their analysis...We find XXX based on our ensemble analysis. These statements might not be generally true but it would add something to the literature.

We are not sure what the reviewer would like us to add. The reviewer should be able to find the estimated median fluxes from the ensemble analysis in our main text (such as line 346). We have already mentioned this information for Northern Asia for instance:

"For Northern Asia (including Eurasia Temperate and Eurasia Boreal), while IS gives large sinks (with an ensemble spread between -2.5 PgC/yr and -0.5 PgC/yr for the whole period), v9 and v7 both show weaker sinks (with a ensemble spread between -1.25 PgC/yr and -0.25 PgC/yr for 2015 and 2016) and a decrease with v9 for 2017 (-0.5 +- 0.5 PgC/yr) and 2018 (-0.25 +- 0.5 PgC/yr).".

5- MAJOR. Line 361: ...we see a trend towards a weaker sink from 2015 to 2018 for northern Asia....Why? The authors should provide some explanation. If they cannot find one then they should admit it.

The trend over Northern Asia has not been investigated and no available reference can provide an explanation for this trend observed in the sink. We have therefore added this sentence page 20 line 395: "Further investigation is needed to explain the decrease in the carbon sink in this region with both IS and v9".

6- MAJOR. Line 368: ...IS seems to follow the pattern of the prior… All priors, some of them? How about the error reduction? Did the authors learn anything from the IS data?

The annual median fluxes for IS appeared to follow the pattern of the prior in 2017 and 2018 over Northern and Southern Tropics. In particular, the median values with the IS ensemble mean give similar values than for the prior, however the ensemble spread is slightly different. When looking at the model individually, the IS does not completely follow the prior of each model (See figure below).

[Figure]

*Fig 3. Monthly mean fluxes in PgC/month (bottom side) and annual mean flux in PgC/yr (top side) of Northern Tropics from 2015 through 2018 among the different models for Prior (left side) and IS (right side).*

However, the goal of the paper was not to give an analysis for each individual model but to give an analysis of the overall average and to compare the v7 MIP and v9 MIP versions.

Additionally, the website provided by the OCO-2 MIP v9 projects shows interactive figures of each model or experience by regions or globally.

If readers would like further information on individual models, the figures are available on this website:

https://gml.noaa.gov/ccgg/OCO2_v9mip/index.php

We have added the link of this website page 35 line 600: "Further information including figures on individual models can be accessed at the following link https://gml.noaa.gov/ccgg/OCO2_v9mip/index.php"

7- MAJOR. Line 375 Fluxes during the recovery period differ between data sources. Why is this important? Any dipoles in neighbouring regions? What about reductions in uncertainties? What have the authors learnt?

Fluxes during the recovery period (2017 and 2018) over the tropics (Northern and Southern Tropics) differ among the data (v9 and IS). This difference seems to come from Northern and Southern Tropical Africa which balance each other, but also from Tropical South America. As we have already mentioned when focusing on these regions: "But this could be also because most of the *in situ* data are located inside the Amazon and not in the Cerrado savanna of Brazil, resulting in IS inversions being dominated by tropical forest seasonality. Alternatively, the OCO-2 inversions could be dominated by savanna seasonality (Baker et al., 2021a in prep)."

Reduction in uncertainties cannot be observed for this period between v7 and v9 as we do not have data with v7 in 2017 and 2018. However, as previously mentioned in the manuscript and for the El Nino period (2015-2016), the reduction in uncertainties are particularly present over smaller regions both in the Tropics or Extra-Tropics. On average, LNLGv9 ensemble spread is smaller than LNv7 and LGv7. Comparing IS and v9, the ensemble spread is, on average, smaller with v9 than with IS, except for Tropical South America where the ensemble spread seems to be significant in this region for both v9 and IS.

8- MAJOR. The discussion is interesting, although this reviewer notes that the authors have made a claim that a previous study (Palmer et al, 2019) using v7 data would probably have been similar using v9 data. And then the authors proceed to compare their results for v9 and those from Palmer et al 2019. This reviewer is unconvinced this is a valid scientific approach. Surely, a cleaner comparison would be to compare their own ensemble values between the two data versions? The discussion about Gloor et al, 2018 and Liu et al, 2017 is a bit odd. What is the authors' point? This

reviewer was also concerned that over the tropics the authors suggest that a good test for fluxes inferred from OCO-2 was fluxes inferred by sparse IS data. That seems like a weak argument. Admittedly, this is a difficult situation (evaluating satellite data using models with poorly characterized errors and sparse in situ data) but in this reviewer's opinion relying on tropical fluxes inferred from IS data is not a great strategy.

The goal of the paper was, as mentioned line 63, to assess if there are any differences between MIPv7 and MIPv9, what would be the implications in the carbon cycle community of using v9 regarding precious studies that used v7. We have been able to observe over the tropics, changes between MIPV7 and MIPv9. Particularly, as observed in figure 9, the differences between the two datasets are not observed for the Northern Tropics but for the Southern Tropics where sources of carbon are observed with v9 while sink of carbon are observed with v7. In our discussion, we are comparing the new results obtained with our v9 ensemble with results provided in the study of Palmer et al., 2019 where they used OCO-2 v7 data in their inversions. Annual flux in PgC/yr are provided in the study of Palmer et al., 2019 allowing us to compare their results with our v9 over the same tropical regions. Our discussion does not mention that this previous study using v7 would have been similar using v9 data. On the contrary, the results of the two studies show that the conclusion of Palmer et al., 2019, in Tropical South Africa, would have been different if they had used v9. Indeed, as we mentioned in our discussion:

"Palmer et al. (2019) assimilated OCO-2 v7 land data and GOSAT v7 data separately during the El Niño period and analyzed the posterior emissions over the pan-Tropical regions. With their inversions, they found carbon sources of 1.56 PgC/yr in 2015 and 1.89 PgC/yr in 2016 over the Northern hemisphere of the tropics. The carbon sources they found with OCO-2 v7 are similar to what we obtain with our OCO-2 v9 inversions. Palmer et al. (2019) found with v7 that the largest seasonal cycle of carbon fluxes in the tropics was over Northern Tropical Africa. This analysis would have probably been similar with v9. However, while they had -0.21 PgC/yr in 2015 and -0.12 PgC/yr in 2016 over the Southern Tropical latitudes, we observe around 0.70 PgC/yr in 2015 and 0.4 PgC/yr in 2016 with v9 (Fig. 9). Analyzing our inversions at regional scales, we saw that this opposite sign in emissions was coming from Tropical South Africa. In our Fig. 11, we have been able to observe a source of around 0.25 PgC/yr in 2015-2016, while v7 emissions were around -0.25 PgC/yr for the same period, which is also what Palmer et al. (2019) found approximately.

They concluded that the largest carbon uptake was over the Northern Congo basin, situated in the Southern Tropical Africa MIP region. This result and the difference between v7 and v9 suggest that the conclusion over the Southern Tropical Africa MIP region with v7 would have been different with v9."

As we mentioned in our paper and summarized in the following table, we can see that the mean values are close between the v7 results of Palmer et al., 2019 and our v7 results:

| | PgC/yr | North Trop. | | North Trop. Africa | | South Trop. | | South Trop. Africa | |
|---|---|---|---|---|---|---|---|---|---|
| | | 2015 | 2016 | 2015 | 2016 | 2015 | 2016 | 2015 | 2016 |
| Mean | Palmer et al., v7 | 1.56 | 1.89 | 1.78 | 1.96 | -0.21 | -0.12 | -0.13 | -0.18 |
| | Crowell et al., 2019 Peiro et al., v7 | 0.89 | 1.21 | 1.18 | 1.33 | -0.06 | -0.2 | -0.21 | -0.18 |
| | Peiro et al., v9 | 1.42 | 1.70 | 1.24 | 1.42 | 0.68 | 0.35 | 0.21 | 0.25 |
| | | | | | | | | | |
| Median | Palmer et al., v7 | 1.55 | 1.64 | 1.89 | 2.02 | -0.26 | -0.05 | -0.06 | -0.20 |
| | Crowell et al., 2019 Peiro et al., v7 | 0.87 | 1.15 | 1.26 | 1.35 | -0.14 | -0.19 | -0.13 | -0.165 |
| | Peiro et al., v9 | 1.40 | 1.42 | 1.26 | 1.36 | 0.70 | 0.38 | 0.18 | 0.27 |

*Table 1: Mean values (top) and median values (bottom) between the ensemble values using OCO-2 v7 from Palmer et al., 2019 and our v7 and v9 results. Yellow colors represent positive fluxes and blue colors represent negative fluxes. Fluxes are in PgC/yr.*

Our IS data also suggests a source of carbon over Southern Tropical Africa with median fluxes like our v9 results. The study of Gloor et al., 2018 took a similar approach to that of Liu et al.,

2017, but instead of using OCO-2 data, they assimilated NOAA surface station network. Our results with v9 and IS are close to those found in Gloor et al., 2018 suggesting a release of carbon over Southern Tropical Africa and particularly near the Congo basin during the 2015 El Nino event. We had to mention this study by Gloor et al., 2018 as they were the first ones to observe sources of carbon in both Northern Tropical and Southern Tropical Africa. And as we also mentioned in our discussion, CO anomalies were found at the same period over this region. Recently, several studies are focusing on Africa to investigate the sources of carbon observed in Tropical Africa.

9- MINOR/MAJOR. Line 384: ...we find better agreement between LNv7 and LNLGv9… So what? Without any context (which this reviewer is sure the authors can provide) this statement about two independent data products is redundant.

Throughout the paper, and when we started looking at latitudinal bands and the MIP regions, we could observe that LNLGv9 is particularly closer to LNv7 than LGv7. As we mentioned in the first notification of this sighting, page 18, line 324: "As we can observe for all other latitudes bands and we will observe for smaller regions, LNLGV9 tends to be closer to LNv7 than to LGv7. This points to previously known issues with the v7 LG data that were resolved with a unified bias correction in OCO-2 v9." This being already explained line 324 and being redundant line 384, we removed the sentence "Here again, we find better agreement between LNv7 and LNLGv9".

10- MINOR/MAJOR. This reviewer was not totally convinced by the authors' use of normalized bias - the MDM has a dynamic range of two orders of magnitude? It would be useful to also report the ensemble and individual model bias as a function of latitude. Similar argument goes for the standard deviation. Appendix?

MDM has a dynamic range of two orders of magnitude. So, the more meaningful interpretation of the residuals comes from the normalized residuals. For instance, a 0.1 ppm residual at the South Pole is more meaningful than a 10 ppm residual at the LEF tower during the summer months.

We have already included the ensemble and individual model bias as a function of latitude for the withheld evaluation since the first submission of our paper. Indeed, figure 12 shows the normalized bias and standard deviation for the ensemble mean of IS and LNLGv9 by latitude. Additionally,

figure 14 shows the normalized bias and standard deviation by latitudes for each individual model. It is not necessary to add them as an annex since they have always been available in the main text.

11- MINOR. TCCON and in situ data are treated as the gold standard. My understanding is TCCON data, despite herculean efforts, still contain biases and those should be acknowledged. Admittedly, those biases are likely smaller than those for OCO-2 but they could be important.

The TCCON data used for validations are average over 30 minutes bins of local solar time and for each site. This gives different bins for different sites, even at the same latitudes or longitudes. TCCON observations are not assimilated but are only used for evaluation purposes, so the uncertainty assigned to the binned observations, which is of 0.5 ppm, does not matter much.

The figure below shows the uncertainty mean (in ppm) over 2015-2018 for each TCCON measurement. We can see that there is large variability in the uncertainties between the different TCCON sites with values between 0.3 ppm and 1.1 ppm. However, as mentioned previously, TCCON uncertainties from the data used in the evaluation are around 0.5 ppm.

[Figure]

*Fig 4. Uncertainty mean for each TCCON measurement over 2015-2018. The uncertainty from the original TCCON data is represented in blue while the uncertainties assigned to the 30 minute bins are orange. TCCON site names are on the left y-axis and latitude in degree corresponding to each site are on the right y-axis.*

12- MINOR. This reviewer was disappointed by the incremental update in the length of analysis compared to Crowell et al 2019. OCO-2 was launched in 2014 and it is still to my knowledge still producing data so why has this study ignored 2019 and 2020? That's 50% more data than they have analyzed! Even if the author took into consideration that they needed 6 months of data at the end of their flux reporting period, they could at least report 2019.

The reviewer and readers need to understand that it takes a year for the modelers to update the MIP protocol and to run the simulations with the new retrievals. Then it takes some months to run the validations of these simulations and some more months to write the paper. This explains the laps between the OCO-2 retrieval versions used and the time the MIP paper is submitted.

Additionally, the incremental update in the length of analysis from our paper is the same length as for Crowell et al., 2019. Crowell et al., 2019 was submitted in 2019 and includes only the years 2015 and 2016. There were hence two years of data, 2017 and 2018, that were not included in the paper. This time range is the same as for our paper. We submitted our paper in 2021 and included four years of data (2015, 2016, 2017 and 2018) and did not include two years of data which are 2019 and 2020.

13- MINOR. On a related note, all the transport models are reasonably well described. However, the description of the inversions is lacking for some models. Uniformity in individual descriptions is needed. For example, what assumptions were made about uncertainties and spatial/temporal correlations - some model descriptions are more comprehensive than others.

We thank the reviewer for this remark where some information was indeed missing for some inversions. We added particularly prior uncertainties for the simulation CT, CSU, and CMS-flux.

The correction can be found on page 38 line 659 for CMS-flux:

 "The prior ocean error is 100%. Fire is not optimized separately; they are part of the NBE."

The correction can be found on page 38 line 675 for CSU:

"Prior ocean comes from a climatology based on Landschutzer v18. Prior standard deviations (independent, no prescribed correlations) for ocean exchange, respiration and GPP were 10% of the net exchange (or respiration or GPP). Fires are from GFED4.1s."

The correction can be found on page 39 line 686 for CT:

"Prior standard deviation is equivalent to 50%. Prior covariance is applied such as the correlations between the same ecosystem types in different Transcom regions decrease exponentially with distance scale $L$ = 2000 km.

More information can be found in:

https://gml.noaa.gov/ccgg/carbontracker/CT2019B_doc.php#tth_sEc8.2"

**References**

Babenhauserheide, A., Basu, S., Houweling, S., Peters, W., and Butz, A.: Comparing the CarbonTracker and TM5-4DVar data assimilation systems for $CO_2$ surface flux inversions, Atmospheric Chem. Phys., 15, 9747–9763, https://doi.org/10.5194/acp-15-9747-2015, 2015.

Chevallier, F., Palmer, P. I., Feng, L., Boesch, H., O'Dell, C. W., and Bousquet, P.: Toward robust and consistent regional $CO_2$ flux estimates from in situ and spaceborne measurements of atmospheric $CO_2$, Geophys. Res. Lett., 41, 1065–1070, https://doi.org/10.1002/2013GL058772, 2014.

Cressie, N., Bertolacci, M., and Zammit-Mangion, A.: From many to one: Consensus inference in a MIP, https://arxiv.org/abs/2107.04208, 2021.

Friedlingstein, P., O'Sullivan, M., Jones, M. W., Andrew, R. M., Hauck, J., Olsen, A., Peters, G. P., Peters, W., Pongratz, J., Sitch, S., Le Quéré, C., Canadell, J. G., Ciais, P., Jackson, R. B., Alin, S., Aragão, L. E. O. C., Arneth, A., Arora, V., Bates, N. R., Becker, M., Benoit-Cattin, A., Bittig, H. C., Bopp, L., Bultan, S., Chandra, N., Chevallier, F., Chini, L. P., Evans, W., Florentie, L., Forster, P. M., Gasser, T., Gehlen, M., Gilfillan, D., Gkritzalis, T., Gregor, L., Gruber, N., Harris, I., Hartung, K., Haverd, V., Houghton, R. A., Ilyina, T., Jain, A. K., Joetzjer, E., Kadono, K., Kato, E., Kitidis, V., Korsbakken, J. I., Landschützer, P., Lefèvre, N., Lenton, A., Lienert, S., Liu, Z., Lombardozzi, D., Marland, G., Metzl, N., Munro, D. R., Nabel, J. E. M. S., Nakaoka, S.-I., Niwa, Y., O'Brien, K., Ono, T., Palmer, P. I., Pierrot, D., Poulter, B., Resplandy, L., Robertson, E., Rödenbeck, C., Schwinger, J., Séférian, R., Skjelvan, I., Smith, A. J. P., Sutton, A. J., Tanhua, T., Tans, P. P., Tian, H., Tilbrook, B., van der Werf, G., Vuichard, N., Walker, A. P., Wanninkhof, R., Watson, A. J., Willis, D., Wiltshire, A. J., Yuan, W., Yue, X., and Zaehle, S.: Global Carbon

Budget 2020, Earth Syst. Sci. Data Discuss., 2020, 1–3, https://doi.org/10.5194/essd-2020-286, 2020.

Gaubert, B., Stephens, B. B., Basu, S., Chevallier, F., Deng, F., Kort, E. A., Patra, P. K., Peters, W., Rödenbeck, C., Saeki, T., Schimel, D., Van der Laan-Luijkx, I., Wofsy, S., and Yin, Y.: Global atmospheric $CO_2$ inverse models converging on neutral tropical land exchange, but disagreeing on fossil fuel and atmospheric growth rate, Biogeosciences, 16, 117–134, https://doi.org/10.5194/bg-16-117-2019, 2019.

Massoud, E. C., Lee, H., Gibson, P. B., Loikith, P., and Waliser, D. E.: Bayesian Model Averaging of Climate Model Projections Constrained by Precipitation Observations over the Contiguous United States, J. Hydrometeorol., 21, 2401–2418, https://doi.org/10.1175/JHM-D-19-0258.1, 2020.

Peylin, P., Law, R. M., Gurney, K. R., Chevallier, F., Jacobson, A. R., Maki, T., Niwa, Y., Patra, P. K., Peters, W., Rayner, P. J., Rödenbeck, C., van der Laan-Luijkx, I. T., and Zhang, X.: Global atmospheric carbon budget: results from an ensemble of atmospheric $CO_2$ inversions, Biogeosciences, 10, 6699–6720, https://doi.org/10.5194/bg-10-6699-2013, 2013.

Saunois, M., Stavert, A. R., Poulter, B., Bousquet, P., Canadell, J. G., Jackson, R. B., Raymond, P. A., Dlugokencky, E. J., Houweling, S., Patra, P. K., Ciais, P., Arora, V. K., Bastviken, D., Bergamaschi, P., Blake, D. R., Brailsford, G., Bruhwiler, L., Carlson, K. M., Carrol, M., Castaldi, S., Chandra, N., Crevoisier, C., Crill, P. M., Covey, K., Curry, C. L., Etiope, G., Frankenberg, C., Gedney, N., Hegglin, M. I., Höglund-Isaksson, L., Hugelius, G., Ishizawa, M., Ito, A., Janssens-Maenhout, G., Jensen, K. M., Joos, F., Kleinen, T., Krummel, P. B., Langenfelds, R. L., Laruelle, G. G., Liu, L., Machida, T., Maksyutov, S., McDonald, K. C., McNorton, J., Miller, P. A., Melton, J. R., Morino, I., Müller, J., Murguia-Flores, F., Naik, V., Niwa, Y., Noce, S., O'Doherty, S., Parker, R. J.,

Peng, C., Peng, S., Peters, G. P., Prigent, C., Prinn, R., Ramonet, M., Regnier, P., Riley, W. J., Rosentreter, J. A., Segers, A., Simpson, I. J., Shi, H., Smith, S. J., Steele, L. P., Thornton, B. F., Tian, H., Tohjima, Y., Tubiello, F. N., Tsuruta, A., Viovy, N., Voulgarakis, A., Weber, T. S., van Weele, M., van der Werf, G. R., Weiss, R. F., Worthy, D., Wunch, D., Yin, Y., Yoshida, Y., Zhang, W., Zhang, Z., Zhao, Y., Zheng, B., Zhu, Q., Zhu, Q., and Zhuang, Q.: The Global Methane Budget

2000--2017, Earth Syst. Sci. Data, 12, 1561–1623, https://doi.org/10.5194/essd-12-1561-2020, 2020.

Schuh, A. E., Jacobson, A. R., Basu, S., Weir, B., Baker, D., Bowman, K., Chevallier, F., Crowell, S., Davis, K. J., Deng, F., Denning, S., Feng, L., Jones, D., Liu, J., and Palmer, P. I.: Quantifying the Impact of Atmospheric Transport Uncertainty on CO2 Surface Flux Estimates, Global Biogeochemical Cycles, 33, 484–500, https://doi.org/10.1029/2018GB006086, 2019

Schuh, A. E., Byrne, B., Jacobson, A. R., Crowell, S. M. R., Deng, F., Baker, D. F.,Weir, B.(2021), Chinese land carbon sink obscured by transport model uncertainty, Arising Matters, (Accepted Reply to Nature Communications, https://doi.org/10.1038/s41586-020-2849-9, not in press yet)

**Reply to Reviewer #2**

The manuscript written by Peiro et al., (2021) is an update of the study made by Crowell et al., (2019), et al., (2019), who assessed the annual and monthly ensemble mean flux derived by nine different global inversions using the assimilation of OCO-2 satellite data (version 7) and in situ data for the period 2015-2016. Helene et al., 2019 used the findings found in Crowell et al. (2019) and compared them with the ensemble median flux of 10 different global inversions based on the assimilation of OCO-2 (version 9) land nadir and land glint (LNLG) data. Further, the authors also analysed fluxes for a more extended period 2015-2018, using both in situ and OCO-2 satellite data separately. In general, the manuscript is well-written, and the discussion and conclusions are interesting. However, some parts in the methodology and results sections need revision and clarifications. I would recommend this paper for publication once the authors have addressed all the questions described below.

We are grateful for the Reviewer comments and for taking the time to review our manuscript. We answered below the comments with information on page and line numbers that have been changed in the manuscript when necessary.

**General comments:**

**1) Line 15.** "However, the lack of data in the tropics limits our conclusions and the estimation of carbon emissions over tropical Africa require further analysis". It would be good If the author can provide spatial maps of LNLG OCO-2 across the world for the studying years to see how bad the coverage is in Tropics. (Maybe show these maps in the appendix of the manuscript).

Thank you for this comment. We took it in consideration and included a map with LNLG OCO-2 10s retrievals in the annex and added a sentence line 268 page 14 "Figure. A1 in the appendix represents the locations of the OCO-2 10s retrievals LNLG for the period of study 2015-2018. We

can see that the posterior flux estimates are constrained with OCO-2 LNLG observations particularly present in the Northern Hemisphere and with a lower number of observations over the tropics."

**2) Line 137.** "This bias correction in v9 allows a more uniformly distribution of $XCO_2$ over regions of interest, decreasing the standard deviation to 0.74 ppm compared to v8, which was of 1.35 ppm." Please indicate that this reduction was found in TCCON Lauder, New Zealand (Fig.13).

We considered this remark and modified the sentence accordingly, line 147, page 7:

"This bias correction in v9 allows a more uniformly distribution of $XCO_2$ over regions of interest, decreasing the standard deviation over the TCCON Lauder (New Zealand) site, for instance, to 0.74 ppm compared to v8 which was of 1.35 ppm."

**3) Line 147-151** Interestingly, the authors accounted for correlations between the model-data mismatch error for the 10 second average of OCO2 soundings. However, the description of how it was calculated is not clear. It would be great if the author could provide more details in this section of the manuscript. Line 159. "Both resulting uncertainty is then passed through the formula to account for error correlation." Which formula? It is hard to understand what the author did to calculate the observation uncertainties. Please provide more details in this section.

We added more information on the error calculation on the 10s average to be more specified of what have been done in reference to the study of Baker et al., 2021. The formula used to account for error correlation was also added in the this section. We consequently changed line 165, page 7 to :

"Since it is known that the uncertainty computed by the retrieval (in variable *xco2_uncertainty*) underestimates the true level of error in the retrieved $XCO_2$, an additional term is added onto this ``theoretical'' uncertainty, in quadrature, to obtain a more realistic uncertainty per scene: $\sigma_{SD}$, the standard deviation of all the $XCO_2$ values used in the 10-second average. In this *ad hoc* approach, scenes that have a very small spread in $XCO_2$ values across the 10-second span are assigned the theoretical uncertainty from the retrieval, while those for which the actual variability of the $XCO_2$

values is larger than the theoretical values are assigned a value closer to this computed error level. Both of these uncertainties are then passed through equation (39) from Baker et al., 2021 to account for error correlations. Finally, an additional error term, $\sigma_{\text{transport}}$, is added in quadrature to account for transport model errors. With all three of these terms considered, the square of the uncertainty on the 10-second $XCO_2$ average is given as:

$$\sigma_{10s}^2 = \frac{1}{\sum \sigma_j^{-2}}\left[(1-c)+c\frac{\left(\sum \sigma_j^{-1}\right)^2}{\sum \sigma_j^{-2}}\right]+\sigma_{SD}^2\left[\frac{(1-c)}{N}+c\right]+\sigma_{transport}^2$$

$\textit{where } \sigma_{SD}^2 = [N\sum X_{CO_{2j}}^2 - \left(\sum X_{CO_{2j}}\right)^2]/N/(N-1),$

and where $XCO_{2j}$ and $\sigma_j$ are the individual $XCO_2$ values going into the average and their retrieval uncertainties, and N is the number of good 10-second $XCO_2$ values in the 10-second average."

**4) Line 160.** "An additional term is added in quadrature to account for transport model errors. This model error term is computed from the difference between the CO2 concentrations computed by the TM5 and GEOS-Chem models when both are driven by the same realistic surface CO2 fluxes, after the annual mean difference field is subtracted off". The author referenced Basu et al., (2018) when they described the transport model error. In Basu et al., (2018), the model part error was calculated by considering the difference between a suite of inverse models optimized with in situ data and OCO-2 retrievals? Did the author do something similar here? Please clarify.

As a remark, the reference Basu et al., 2018 does not refer to the model error term but to the fact that the uncertainties between different 10s averages are assumed to be independent when assimilated. Here is the full text : *"Finally, an additional term is added in quadrature to account for transport model errors. This model error term is computed from the difference between the CO_2 concentrations computed by the TM5 and GEOS-Chem models when both are driven by the same realistic surface CO_2 fluxes, after the annual mean difference field is subtracted off; the values that result are considerably smaller than those model errors added on for the OCO-2 v7 MIP (Crowell et al. 2019). In contrast to this level of detail, the errors between different 10s averages are assumed to be independent when assimilated into the inversions (Worden et al., 2017; Crowell et al., 2019). Several studies have used this method in order to be coherent with the*

*resolution of their inversions or simulations regarding the OCO-2 resolution (Basu et al., 2018; Chevallier et al., 2019)."*

In order to prevent confusion regarding which of the two separate points the reference refers to, we have changed the last sentence to: "Several studies have used this assumption, deeming it appropriate for the resolution of their inversions or simulations (Basu et al., 2018; Chevallier et al., 2019)".

However, to give a specific answer to the question, the model error that we use here is substantially different from that calculated in Basu et al. (2019). In that reference, the difference between the modeled $CO_2$ fields reflects the full impact of model differences on the inversion process as a whole, instead of just on a forward model run, since each model started by inverting the same set of in situ data. Other factors, such as the sparsity and distribution of the data, and the data uncertainties assumed, could contribute to the spread in the model $CO_2$ fields they obtained. In contrast, here we isolate pure transport model differences by running an identical set of $CO_2$ fluxes forward through two different models which are TM5 and GEOS-Chem: the difference that result is due only to differences in the models' transport (and other model features, such as resolution). The fluxes used were realistic insofar as they were taken from an assimilation of in situ $CO_2$ data (from CarbonTracker, 2017, Peters et al., 2007, with updates documented at http://carbontracker.noaa.gov). The resulting $CO_2$ mixing ratios were sampled at the early afternoon OCO-2 overflight times and a multi-year seasonally-varying climatology was developed. The deviations from this climatology from year to year were examined; and the time-varying standard deviations obtained were then used as model errors in the inversions.

**5) Line 240.** For LNLGv9 inversions, the available 10s OCO-2 retrievals were averaged and compared to TCCON observations". I think the author means prior and posterior $XCO_2$ simulated retrievals were averaged and compared to TCCON?

Indeed. We modified this sentence by adding line 256: "For LNLGv9 inversions, the available 10s prior and posterior $XCO_2$ simulated retrievals were averaged and compared to TCCON observations […]."

**6) Line 257-258.** Please indicates that Figure 5 represent only the median annual flux and the monthly median flux. In Crowell et al., 2019, the authors discussed the annual and monthly mean flux, not the median. Could you indicate why you decided to compare the median and not the mean of the flux?

The use of median annual flux and monthly median fluxes has been added in the sentence page 15 line 293: "Figure 6 represents the median annual emissions (in PgC/yr, for the left panels) and the monthly median emissions (in PgC/month for the right panels) at the global scale for each experiment." We decided to use the median flux instead of the mean flux in order to visualize fluxes values given by most of the models. Some models might be biased for some regions for instance. We were particularly interested to not consider these outliers models and therefore to consider only the median fluxes.

**7) Line 261-262:** "However, the peak sinks during the Northern Hemisphere growing season (from May through September) are slightly larger with OCO-2 v7 than with OCO-2 v9". Something is missing here. In this section, you discuss global flux estimates, and at the same time, the author discusses fluxes in the Northern hemisphere. I cannot see the Northern Hemisphere Figure for this section.

In this figure, we are indeed representing the $CO_2$ fluxes at global scale. But the terrestrial land sink during the Northern Hemisphere growing season (from May through September) has an impact at global scale that we can observe on this figure (global scale). Impact on the seasonal cycle that have been also observed in Crowell et al., 2019 with the v7 MIP. Other previous studies also mentioned the sink during the growing season for Northern land regions (Baker et al., 2006). Byrne et al., (2017) also observed that the GOSAT nadir land data has a large sink at global scale in June-July-August due to the strong biospheric uptake of $CO_2$ during the Northern Hemisphere growing season. Finally, the recent study of the Global Carbon Budget 2020 by Friedlingstein et al., (2020), mentioned that the terrestrial land sink during May-Jun-July-August could be due to the combined effects of two reasons : the rise of plant growth fertilizer that increase the imput of $CO_2$ ; and the lengthening of the growing season in the Northern temperate and boreal areas.

In the case that other readers do not know the impact of Northern Hemisphere growing season at global scale, we've added page15 line 283 "The large sink observed at global scale during this season is due to the strong biospheric uptake of CO2 in the temperate and boreal forest of Northern Hemisphere (Friedlingstein et al., 2020)".

**8)** The author includes LoFI inversion in the paper. However, there are no comments about their results. For example, LoFI seems to agree with the prior median estimate over the Northern Extra Tropics but not with IS and LNLGv9 assimilated flux.

LoFI was part of the v9 MIP not as a standard inversion but as an additional metric of flux inversion. Our study was not focusing at evaluating LoFI and comparing it with the other inversions. The goal of our paper was to compare the v7 MIP with the v9 MIP results. Consequently, we added additional information on LoFI line 82 page 4, to better explain its difference and why it was initially used in the v9 MIP : "The LoFI submission (Weir et al., 2021), new in the v9 MIP, is intended as an additional metric of flux inversion skill. LoFI uses in situ observations to match only the global atmospheric growth rate with an empirically derived land sink (Chevallier et al., 2009). The inferred fluxes are thus independent of the spatial and sub-annual variability in atmospheric observations and rely minimally, if at all, on model atmospheric transport representation. Despite the weak data constraint, it is included below with the IS inversions because it depends on the annual, global growth rate determined from observations. Given the problems flux inversions have facing remote-sensing retrieval biases (O'Dell et al., 2018) and atmospheric transport errors (Schuh et al., 2019), LoFI serves as a first-order check on inversion skill. Times and places where a flux inversion outperforms or equals LoFI's skill suggest a nominally operating system, while significantly degraded skill suggest a problem, e.g., in the prior, atmospheric transport, and/or ingested data."

**9) Line 310- 313.** "In the Southern Extra-Tropics, the authors mention that the sink flux estimate with OCO-2v9 is stronger than estimates made with IS, and v7, mainly because the ensemble spread with v9 is larger than with v7." I am a bit surprised that fluxes assimilated with v9 have a larger spread than v7.

OCO-2 v9 data supposed to have lower biases than v7. It seems that the error in the transport model might be the consequence of the large spread in the inversion with v9.

As a remark, and to include exactly what we have written in our paper, here the paragraph :

*" They show, for the whole period, stronger sinks with v9 than with IS, and v7. However, in contrast to NHExt, the ensemble spread is larger with v9 than with v7. The bias reduction of v9 gives a smaller spread and hence a better agreement among the models, particularly over the Northern Hemisphere.*"

We do not justify the stronger sink in the Southern Hemisphere because there is a larger spread among the models with v9 than with v7 in this latitudinal band. However, we do mention that the ensemble spread among the model seems to be larger over the Southern Hemisphere with v9 than with v7.

As mentioned previously for the transport model error, the error in the transport model is calculated using the differences between TM5 and GEOS-Chem and added in the uncertainties. As mentioned, the difference among the models is mainly due to differences in the models' transport. Furthermore, and compared to the v7 MIP, there are more common elements across the models in the v9 MIP where all modelers used the same *in situ* data as well as the same errors on the OCO-2 10s averages.

Over the Southern Hemisphere, there is less land cover than in the Northern Hemisphere, so we know that there are few land retrievals to constrain the land fluxes. The ensemble spread could reflect the significant uncertainty of land fluxes in this region. It could also reflect the bias in satellite retrievals due to large solar zenith angles at this latitudinal band, which indicate that there is not enough sunlight available to retrieve data. The spread between models is data-driven and reflects the models' abilities to simulate observations. All these elements could explain the lack of concordance between the models, observed mainly for this region with v9.

**10) Line 314-318.** Over the Tropics, the authors mention that posterior flux estimate (LNLG) is quite different to the prior estimate (Fig 6.c), but not comments are provided. How reliable can be the posterior estimate over the Tropics knowing that OCO-2 retrievals can be biased due to cloud

coverage during the wet season and aerosol from biomass burning during the dry season, as the author mentioned?

The lack of validation data over the tropics makes this comment difficult to answer. As we mentioned in the paper, there is more OCO-2 data in tropics than the IS data but the OCO-2 data are biased during the dry season due to aerosol from biomass burning and due to cloud during the wet season. There is not enough validation data as well to estimate which from the OCO-2 or the IS posterior emissions are more reliable. Some aircraft project efforts organized by the National Oceanic and Atmospheric Administration are ongoing to address these biases.

**11)** Could you explain a bit more about the dipole in northern Africa? You mentioned several studies, but a better explanation is needed.

In order to give more details about this dipole between Europe and Northern Africa we changed our sentence page 20 line 373 :

"Previous studies already observed and mentioned a larger European land sink in balance with a large tropical land source. Particularly, Houweling et al. (2015) found a difference in flux between these two regions of around 0.8 PgC/yr. They found that this balance was caused by a lack of GOSAT observations during the winter over Europe. Additionally, Chevallier et al. (2014) also observed this balance between Europe and Northern tropical Africa in their GOSAT inversions and they considered the large source over North Africa has unrealistic. According to Feng et al. (2016) the large sink over Europe inferred from GOSAT data was caused by large biases outside of the region, which for mass balance, the inversions was removing larger $CO_2$ over Europe, in agreement with Reuter et al. (2014) and Reuter et al. (2017)."

**12)** Fig 8 (NH Tropics) and (SH Tropics) LoFi monthly seasonally seem to be offset? Could the author explain why LoFi might not be capturing the seasonally over these latitudes bands?

As mentioned above with more detail for the comment (9), LoFI uses a different method than the other inversions but fit some independent data as the other simulation do. LoFI has then been used in this MIP project to look at a range of different methods.

LoFI is not considered as a standard inversion compared to the other models, it is a bit like an outlier. And the comparisons to *in situ* data (Figure 11) show that there's probably some degraded skill for LoFI in the tropics, but not much due to the significant uncertainties there. In the Fig. 3 of Weir et al., 2021, LoFI is compared to inversions with a broader collection of priors than used in the MIP. It does have a bit of a phase shift compared to these inversions but it is within the uncertainty range when compared to all the TRENDY v7 models.

**13) Line 412.** Alternatively, the OCO-2 inversions could be dominated by savanna seasonality (Baker et al., 2021a in prep). I cannot find (Baker et al., 2021a) in the reference list, and the reference cited (Baker et al., 2021) does not mention anything about the savanna seasonality but how to calculate the error correlations in OCO-2 data. Please provide the correct reference.

This referred to a paper in preparation by Ian Baker. The paper has not been submitted yet. We removed the year in the citation and added the first name of the author for no confusion with the paper of David Baker et al., 2021.

**14) Line 426-.** "Over the southern hemisphere, a large underestimation of the ensemble mean of LNLGv9 appears compared to the observations". I am surprised with these findings; I would have thought that the spatial coverage that OCO-2 data in the southern hemisphere would improve the results compared to IS. I am also surprised by the significant difference between LNLGv9 and IS biases in SH. Do you know how large are LNLGv9 OCO-2 biases in SH compared to v7? You mentioned in the introduction that biases in LNLG v9 were reduced considerably compared to v8 and v7.

Figure 12 shows the normalized bias and standard deviation for the ensemble mean of IS and LNLGv9. For the biases evaluation with the withheld data, we can see an increase in the biases going from the Northern latitudes (of maximum 0.5) to the Southern ones (reaching -3.5 for the

LNLGv9). The variability seems to follow as well, where the variability is larger over the Southern Hemisphere than the Northern Hemisphere. This shift of variability and biases between the Northern and Southern Hemispheres seems to be linked with the number of withheld data. We can see Figure 2 that the number of withheld data is lower over the Southern Hemisphere with less than 1000 data, while the number of observations is above 10 000 for the Northern Hemisphere. This large underestimation observed with LNLGv9 over Southern Hemisphere when compared to the withheld data could be explained with the lack of withheld data over this Hemisphere.

We added this sentence page 25 line 464 : "In addition, this same variability seems to be disproportional to the number of withheld data (see Fig. 2). Indeed, the standard deviation for both IS and LNLG is low when the number of withheld data is important (superior to 10 000 data)."

Furthermore, the MDM values and withheld data were not considered during the v7 MIP. This attempt to account for correlations between the MDM errors was made only for the v9 MIP. Evaluating v7 with the withheld data cannot be performed. However, the biases in LNLGv9 have been reduced considerably compared with v8 and v7 when evaluated with the TCCON data as discussed in the TCCON section (3.4.3).

**15)** Is it possible to provide a Table in the appendix with no normalized bias?

We thought more relevant to use normalized values for this evaluation as the ranges of data among the models can be large and different. Particularly, since the MDM values range over two orders of magnitude, the use of the normalized residuals gives the most meaningful interpretation of the residuals. We decided then to keep the normalized values for better comparison of scale variability. Additionally, more information on the scaling can be found with the RMSE (root-mean-square error) values in ppm. Using RMSE we can estimate how much the models differ from the withheld data.

**16) Line 466.** It is possible then that this excess of concentration, in both experiments, reflects the initial conditions of the inversion. Is there any way to test this?

One possible way to verify this would be to have vertical profile of convective mass fluxes comparing the meteorological conditions first (such as the convection from ERA-Int, GEOS-FP or MERRA-2 for instance). Schuh et al., 2019 suggested that the differences and biases observed between TM5 and GEOS- Chem could result from the representation of vertical motion. Ongoing studies are focusing on this difference, such as Schuh et al., (in prep) where their new finding suggest difference mainly coming from the meteorological input.

**17) Fig 13 and Fig 14.** I don't understand why the author normalized the biases. It is clear that RMSE is large at some latitudinal bands and TCCON sites, suggesting that raw biases might also be high. I would also consider adding this information to the manuscript.

We are not sure to really understand what the reviewer means for this comment regarding the TCCON sites. Figure. 13 and Fig. 14 are showing normalized biases for the withheld data and we justified in the previous comment (16) and in the main text (line 470 page 28) why we are using the normalized biases when evaluating with the withheld data.

**18) Line 471-474.** "Compared to the evaluation of ISv7 in the study of Crowell et al. (2019), ISv9 and LNLGv9 biases are closer to each other (in the v7 MIP, the LNv7 were biased high compared to ISv7). Additionally, the OCO-2 biases have decreased (to values between -1.0 and 1.0 ppm) with v9 compared to v7, where biases ranged between -1.5 and 1.5 ppm (Crowell et al., 2019)". I don't understand why the authors say that OCO-2 biases have decreased compared to Crowell et al., 2019. Did Crowell et al., 2019 normalize the bias? If not, I don't quite understand the comparison.

As comment, the biases are normalized only for the withheld evaluation, they are not normalized for the TCCON evaluation. This paragraph refers to the TCCON evaluation and to figure 17 where the biases are not normalized. We then were able to compare our TCCON evaluation with the one used in Crowell et al., 2019.

**19) Line 474-476.** "…to the accuracy of TCCON retrievals over these regions (Crowell et al., 2019). I think that biases might likely be associated with biases to satellite observations than TCCON bias. Besides, I dont think that Crowell et al., 2019 is not a good reference for talking about the accuracy of TCCON. XCO2 TCCON retrievals can contain airmass-dependent biases, which are corrected using the method described by Wunch et at.. Just wondering how wrong could be this correction to cause the posterior concentration biases seen here.

Even if this has been mentioned in Crowell et al., 2019, we removed this comment in order to give two assumptions for this positive bias observed over most of the European sites. We changed consequently line 507 page 32 : "As observed here as well, IS and v9 have large positive biases over most of the European sites, which could indicate either an issue related to the coarse resolution used by the transport models or to a latitudinal bias (though this is not shown here, positive biases are also observed for the East Trout Lake TCCON site situated in Canada at almost the same latitudinal band as the European sites)".

**20) Line 476.** Could the authors provide the location of the TCCON sites in a map instead of Table 3? It would be better for the reader to see where Caltech, Saga and Tenerife TCCON sites are located. I had to look at their locations from other manuscripts.

We provided a map of the TCCON sites location and included it page 13 with the sentence line 258 page 13 "All TCCON sites used in the evaluation section are listed in Table 3 and Fig. 3 represents the location of the TCCON sites".

**21)** The Caltech, Saga, and Tenerife sites show large underestimations in the IS and v9 results across all models. Isn't it bad to find a large underestimation at Caltech for the reliability of the fluxes estimated in North America? Is the location of Caltech a coastal site in the model? If so, it might strongly affect by ocean fluxes where not OCO-2 observations were assimilated.

As mentioned in our paper, Caltech shows indeed large underestimation in the IS and v9 results across all models, however Edwards site which is very close to Caltech shows lower bias with either underestimation or overestimation according to the models used. Caltech is not the only one

site representative of the Northern America however. Lamont situated in Oklahoma and Park Fall in Wisconsin, are also part of the Northern America TCCON sites.

Regarding the comment that Caltech might be affected by ocean fluxes where no OCO-2 data are assimilated, this might not be a reason for the underestimation observed with IS and LNLG simulations. Indeed, when we look other coastal TCCON sites like Rikubetsu, Tsukuba or Ascension Island, they do not show large underestimation but a slightly overestimation. We discuss about these other sites in the paper. Besides, as we already mentioned in the paper and observed in v7 MIP as well, differences between the Caltech and Edwards sites (which are very close each other) could be due to the location of Edwards over the mountains while Caltech is affected by the Los Angeles basin (Kort et al., 2012; Schwandner et al., 2017). So, the coarse resolution of models cannot differentiate the variability of these two sites (Crowell et al., 2019; Schuh et al., 2021).

**22) Line 480.** Another possible explanation of the underestimation observed over Saga and Izaga is that these small islands are strongly influenced by ocean fluxes, where the assumed uncertainties are small compared to land.

We thank the reviewer for this comment. This underestimation could indeed be linked to the fact that they are small islands. But the bias observed in Izana could also be linked to the high altitude of the TCCON site. We modified the paragraph, line 514 page 32, by :

"This could also explain the underestimation observed over Saga and Paris, which are urban regions. However, Saga is also a small island and could hence be influenced by ocean fluxes, where the assumed uncertainties are small compared to land. The underestimation observed for Izana (Tenerife Island) is probably linked to the same uncertainty (being a small island) but could also be due to the high altitude of the site."

The biases in v9 have decreased for Ascension Island compared to v7, where the biases were around 1.0 ppm for LN and LG. Are these results compared to TCCON biases presented in Crowell 2019? If so, I don't understand why the authors compare standardized bias against not standardized biases?

In our v9 MIP evaluation with the TCCON data, we presented bias and standard deviation for all TCCON sites by model. All evaluations in our paper are only for the v9 simulations. We did not include v7 simulations as the evaluation of the v7 MIP is already presented in the paper of Crowell et al., 2019. We do not understand why the reviewer mentioned that a comparison between standardized bias against not standardized biases has been done in the paper when it is not the case. The way we did our evaluation with TCCON is similar to what has been done in the paper of Crowell et al., 2019.

As a remind also, biases were only normalized for the withheld evaluation.

**23) Line 495.** "Transport model uncertainty is not expected to have changed dramatically since v7. This suggests that the reduction in the ensemble spread is likely related to a decrease in OCO-2 retrievals errors in v9 compared to v7". Are the modellers that participate in OCO-2 MIP have trying to consider improvement in the transport modelling (at the surface)?

We thank the reviewer for this comment. All of modelers did not modify or updated their transport model between the two MIP versions, except the simulation CAMS. The transport model used in CAMS is different between the two versions. The meteorology has been updated with ERA5 instead of ERA-Int. Then, the vertical mixing is different between the two versions where it has been changed from Tiedtke., (1989) to Emanuel., (1991) convection scheme with addition of thermal terms. But further work is needed to demonstrate which of transport version performs the best.

Modelers participating in the OCO-2 MIP are involved in efforts to discern the *causes* for differences in the two most common CTMs used for the MIP, TM5 and GEOS-Chem. Experiments, and corresponding papers (Schuh et al., in prep) are underway focusing on differences in parameterized vertical transport. This work is still at the point of discerning specific causes for differences, as opposed to fixing any known issues with either model. Therefore, we are left with the assumption, for the time being, that the differences between the CTMs, including near surface behavior as pointed out, arise primarily as differences in the parent model meteorology that drive both models.

**24) Line 525-526.** "When they compared..." who is they? I think the reference is missed here.

"They" was referring to the study of Gloor et al., 2018. We added this reference in the sentence.

In the discussion section, I think it is also important that the author indicates why the assimilating OCO2 LNLGv9 data over the Tropics shows a stronger seasonality than prior fluxes. For the reader, it would be good to know why prior fluxes derived from ORCHIDDE or CARDAMON are not capturing well the seasonality. What could be wrong with these models?

Figure 9.b and 9.d shows the monthly median fluxes for the different simulations and prior over Northern Tropics and Southern Tropics respectively. We can see on this figure that the prior (in black) has the same seasonality than the OCO-2 LNLGv9 (in dark blue) but a difference in magnitude. The sources are particularly lower with the priors than with the LNLGv9. For information regarding the priors used, only two models used ORCHIDDE or CARDAMON, the rest used CASA-GFED.

Net fluxes in the tropics are sensitive to the Gross Primary Productivity (GPP) and the Respiration (R). However, large fluxes are present in the tropics which can give large GPP and R in this region. Consequently, the way they are calculated in these models can bring not negligible differences between the priors and the posteriors. Additionally, these two lands models are parameterized mainly with Northern Hemispheric information and so might not represent correctly the tropical latitudes (Ian Baker et al., in preparation). More information on the ORCHIDDE, CARDAMON and CASA-GFED comparison will be provided in this future paper, which was not the subject of our paper.

**25) Line 552 -554** "…with a slight negative bias in the v9 OCO-2 data for almost all latitudes, particularly in the Southern Hemisphere and the tropics, where few evaluation data are available". Here the author mentions "slight negative bias in the v9 OCO-2 data", however in lines 426-427, they write, "Over the southern hemisphere, a large underestimation of the ensemble mean of LNLGv9 appears compared to the observations". This seems a bit contradictory.

The results, in the evaluation section using the withheld data, show large negative bias in the Southern Hemisphere but slightly negative over the tropics. We modified line 589 with :

"particularly large in the Southern Hemisphere and slightly negative in the tropics,"

**Editorial and minor comments:**

**Title:** Four years of global carbon cycle observed from OCO-2 version 9 and in situ data, and comparison to OCO-2 v7. As a personal opinion, I would write version 7 instead of v7 to be consistent with "version 9".

We considered this remark and changed the title.

**Line 121.** "Inversions assimilating OCO-2 ocean retrievals produced unrealistic results with annual global ocean sinks higher of $2.6 \pm 0.5$ GtC.yr$-1$ compared to the state-of-the-art estimated in Le Quéré et al. (2018)." Could you quote the ocean sink estimated in Le Quéré et al. (2018).

The ocean sink estimated in Le Quéré et al. (2018) has been added in the sentence page 7 line 134: "inversions assimilating OCO-2 ocean retrievals produced unrealistic results with annual global ocean sinks higher of $2.6 \pm 0.5$ GtC/yr compared to the state-of-the-art estimated in Le Quéré et al. (2018), which was of $2.5 \pm 0.5$ GtC/yr in 2017."

**Line 136**. (Kiel et al. (2019)). Remove parenthesis to 2019. We removed the parenthesis.

**Line 151.** "Details of the form and derivation of these average uncertainties may be found in the 'constant correlation' section of Baker et al. (2021).", Please be consistent with the quotation mark "constant correlation". We changed the quotation mark.

**Line 137.** "This bias correction in v9 allows a more uniformly. Replace the word uniformly by uniform. We replaced it.

**Line 184.** There is an extra point. Please remove. **Extra point removed.**

**Line 241-242 Figure3.** Please add the y-axis description (right legend). **The y-axis description has been added.**

**Line 311.** "However, in contrast to NHExt". In this line, you need to define define NHEx.

We added it page 18 line 335 : "However, in contrast to the Northern Extra-tropics (NHExt)"

**Line 305:** The Southern extra-tropics. Capital letter?. **We changed it as well as everywhere else in the text.**

**Line 306.** Correct the word "signifcaintly" to significantly. **We modified it.**

**Line 398.** For the monthly emissions of the Southern Tropical regions (Fig. 10), we can see the strong impact of El Niño in fall 2015 over Southern Tropical Asia in the larger emissions (with a maximum of 0.35 ±0.01 PgC/yr) given by all the inversions compared to the rest of the period. Please re-write, it is hard to understand.

We modified this sentence by page 23 line 430:

"Looking at the monthly emissions of the Southern Tropical regions (Fig. 11), we can see the strong impact of El Niño between August and November 2015 over Southern Tropical Asia. The emissions reach a maximum of 0.35 ±0.01 PgC/yr, highest of around 0.30 PgC/yr compared to the rest of the period."

**Line 400.** This large fall 2015 mainly come from Indonesian fires. I would write: The large emissions from Southern Tropical Asia (Fig.10.f) primarily come from Indonesia fires. I would also remove the cyan and green lines from all plots (Fig.5 to Fig.10) that show monthly median

fluxes for 2018-2019. If there is no data there, I don't think it is needed to show that the median is zero.

We considered this suggestion and modified the sentence accordingly. We also changed all figures in order to remove the median values when they are equal to zero.

**Line 605.** … is about 0.5 GtC.yr−1.Land (need space) We added a space.

**References :**

Baker, D. F., et al. (2006), TransCom 3 inversion intercomparison: Impact of transport model errors on the interannual variability of regional CO2 fluxes, 1988–2003, Global Biogeochem. Cycles, 20, GB1002, doi:10.1029/2004GB002439.

Byrne, B., D. B. A.Jones, K. Strong, Z.-C. Zeng, F. Deng, and J. Liu (2017), Sensitivity of CO2 surface flux constraints to observational coverage, J. Geophys. Res. Atmos., 122, 6672–6694, doi:10.1002/2016JD026164. Received

Friedlingstein, P., et al. (2020), Global Carbon Budget 2020, Earth System Science Data. Issue 12, Pages 3269-3340, doi.org/10.5194/essd-12-3269-2020.

Tiedtke, M., 1989: A comprehensive mass flux scheme for cumulus parameterization in large-scale models. *Mon.Wea.Rev.,* 17, 1779-1800, doi:10.1175/1520-0493(1989)117

Emanuel, K. A., 1991: A scheme for representing cumulus convection in large-scale models. *J. Atmos. Sci.*, **48**, 2313–2329, doi:10.1175/1520-0469(1991)048<2313:ASFRCC>2.0.CO;2.